# Genetic code expansion, click chemistry, and light-activated PI3K reveal details of membrane protein trafficking downstream of receptor tyrosine kinases

Duk-Su Koh[1†], Anastasiia Stratiievska[1†], Subhashis Jana[2], Shauna C Otto[1], Teresa M Swanson[1], Anthony Nhim[1], Sara Carlson[1‡], Marium Raza[1], Ligia Araujo Naves[1§], Eric N Senning[3], Ryan A Mehl[2], Sharona E Gordon[1*]

[1]University of Washington, Department of Physiology & Biophysics, Seattle, United States; [2]Department of Biochemistry and Biophysics, Oregon State University, Corvallis, United States; [3]Department of Neuroscience, University of Texas at Austin, Austin, United States

*For correspondence: seg@uw.edu

†These authors contributed equally to this work

Present address: ‡Allen Institute for Cell Sciences, Seattle, United States; §Instituto de Ciências Biológicas, Universidade Federal de Minas Gerais, Belo Horizonte, Brazil

Competing interest: The authors declare that no competing interests exist.

**Abstract** Ligands such as insulin, epidermal growth factor, platelet-derived growth factor, and nerve growth factor (NGF) initiate signals at the cell membrane by binding to receptor tyrosine kinases (RTKs). Along with G-protein-coupled receptors, RTKs are the main platforms for transducing extracellular signals into intracellular signals. Studying RTK signaling has been a challenge, however, due to the multiple signaling pathways to which RTKs typically are coupled, including MAP/ERK, PLCγ, and Class 1A phosphoinositide 3-kinases (PI3K). The multi-pronged RTK signaling has been a barrier to isolating the effects of any one downstream pathway. Here, we used optogenetic activation of PI3K to decouple its activation from other RTK signaling pathways. In this context, we used genetic code expansion to introduce a click chemistry noncanonical amino acid into the extracellular side of membrane proteins. Applying a cell-impermeant click chemistry fluorophore allowed us to visualize delivery of membrane proteins to the plasma membrane in real time. Using these approaches, we demonstrate that activation of PI3K, without activating other pathways downstream of RTK signaling, is sufficient to traffic the TRPV1 ion channels and insulin receptors to the plasma membrane.

## eLife assessment

This study develops a new and **important** method for dissecting out two overlapping cell signaling pathways, phosphoinositide signaling and membrane protein trafficking. The combination of two state-of-the-art spectroscopic techniques provides **compelling** evidence for a reciprocal influence between an enzyme and a channel. The work will be of interest to the broader cell biology, biophysics and biochemistry communities.

## Introduction

### RTKs activate phosphoinositide 3-kinase

Receptor tyrosine kinases (RTKs) are the second largest class of cell surface receptors, with 58 members in the human genome (*Robinson et al., 2000*). Binding of agonists, typically polypeptides such as growth factors, to their extracellular amino-terminal domains results in autophosphorylation of their intracellular carboxy-terminal domains (*Lemmon and Schlessinger, 2010*). The autophosphorylated

receptors then signal to activate phospholipase Cγ, mitogen-activated protein kinase, and phospho-inositide 3-kinase (PI3K). The complex web of signals downstream of RTKs underlies their importance in a cell regulation and dysregulation but also makes studying the roles and dynamics of any particular branch of their signaling pathways challenging.

The Class IA PI3K coupled to RTKs are obligate heterodimers composed of regulatory p85 and catalytic p110 subunits (*Figure 1A*; *Geering et al., 2007*). The p85 regulatory subunit contains two SH2 domains (N-SH2 and C-SH2) separated by an inter-SH2 (iSH2) domain. The p110 catalytic subunit interacts with the entire N-SH2-iSH2-C-SH2 region of PI3K, with the N-SH2 acting as an autoinhibitory domain to inhibit catalysis. For the well studying RTK tropomyosin receptor kinase A (TrkA), binding of its agonist nerve growth factor (NGF) triggers autophosphorylation of a pair of tyrosines. Phospho-TrkA then binds to the PI3K N-SH2. This has the dual effect of recruiting PI3K to the PM (*Thorpe et al., 2015*; *Ziemba et al., 2016*) and causing a conformational change in p85 to relieve autoinhibition of p110 by the N-SH2 (*Zhang et al., 2020*). Activation of the RTK insulin receptor (InsR) by its agonist, insulin, also involves autophosphorylation of tyrosines on its intracellular domain and subsequent activation of PI3K; however, in the case of InsR, various insulin receptor substrate adaptor proteins link InsR to PI3K (*Ye et al., 2017*; *Saltiel, 2021*). PI3K bound directly or indirectly to activated RTKs on the membrane then phosphorylates phosphoinositide 4,5-bisphosphate ($PI(4,5)P_2$) to generate phospho-inositide 3,4,5-trisphosphate ($PI(3,4,5)P_3$). $PI(3,4,5)P_3$ is a well-recognized signal for membrane fusion in organisms ranging from *Dictyostelium* (*Nichols et al., 2015*) to humans (*Hawkins et al., 2006*; *Hawkins and Stephens, 2015*). Indeed, $PI(3,4,5)P_3$-triggered membrane fusion mediates tumor cell motility, with PI3K acting as an oncoprotein and its corresponding phosphatase, PTEN, as a tumor suppressor (*Hawkins et al., 2006*).

## RTK/PI3K signaling increases sensitivity to painful stimuli via TRPV1 ion channels

Increased sensitivity to painful stimuli in injured and inflamed tissue is due to adaptations in the peripheral and central nervous systems (*McMahan et al., 2013*). For peripheral sensitization, G-protein-coupled receptors and RTKs signal through multiple pathways to produce cellular changes across multiple time scales. NGF is an inflammatory mediator released from leukocytes, including mast cells, eosinophils, macrophages, and lymphocytes, in response to tissue injury and inflammation (*Freund-Michel and Frossard, 2008*; *Reis et al., 2023*). Paradoxically, in addition to its role in increasing sensitivity to painful stimuli, NGF is also critical for wound healing (*Ebadi et al., 1997*; *Lambiase et al., 2000*; *Bonini et al., 2002*). Thus, strategies to block NGF-mediated pain sensitization have the unfortunate consequence of interfering with post-injury recovery.

One of the mechanisms by which NGF increases sensitivity to pain involves a PI3K-induced increase in the number of pain-transducing TRPV1 ion channels in the plasma membrane (PM) of sensory neurons (*Bonnington and McNaughton, 2003*; *Stein et al., 2006*; *Zhu and Oxford, 2007*). We recently discovered reciprocal regulation between TRPV1 and PI3K. We found that the NGF-stimulated rise in the PI3K product $PI(3,4,5)P_3$ is much greater in cells transiently transfected with TRPV1 than in control cells without TRPV1, i.e., TRPV1 enhances NGF-mediated PI3K activity (*Stratiievska et al., 2018*). Interestingly, TRPV1 interacts directly with PI3K (p85α and p85β subunits; *Figure 1A*) via its amino-terminal ankyrin repeat domain (ARD) (*Stein et al., 2006*) and the ARD is sufficient to increase NGF-induced PI3K activity (*Stratiievska et al., 2018*). Based on these findings, we propose that TRPV1 regulates NGF-induced PI3K activity via this direct interaction.

## Insulin/PI3K signaling

Activation of InsR by insulin signals via PI3K to activate a powerful protein kinase, Akt, which in turn regulates multiple cellular processes involved in metabolism (*Saltiel, 2021*). The PI3K/Akt pathway is thus of great interest in understanding obesity and type 2 diabetes. Insulin-dependent recycling of InsR has also been extensively studied (*Knutson, 1991*; *Chen et al., 2019*). Ligand-simulated InsR endocytosis is followed by either targeting to lysosomes for degradation or recycling endosomes for reinsertion into the PM. This return to the PM is not fully understood, but it is believed to involve small Rab family GTPases (*Iraburu et al., 2021*). Importantly, pharmacological inhibition of PI3K has been shown to inhibit InsR trafficking to the PM, indicating that PI3K activity is necessary for InsR homeostasis (*Sasaoka et al., 1999*).

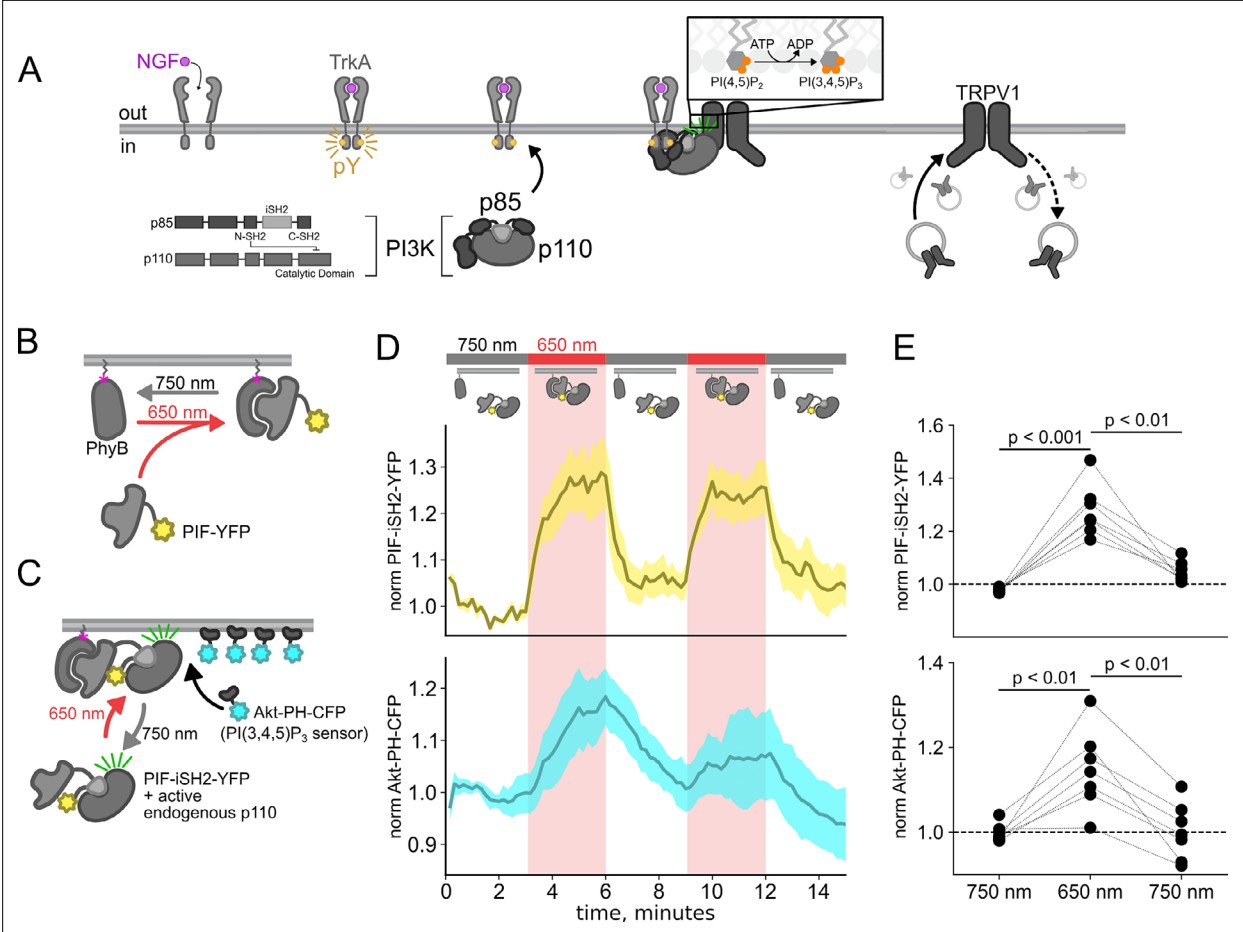

**Figure 1.** Using the PhyB/PIF system to activate phosphoinositide 3-kinases (PI3K) with light. The OptoPI3K system reversibly activates PI3K to generate phosphoinositide 3,4,5-trisphosphate (PI(3,4,5)$P_3$) at the plasma membrane (PM). (**A**) Diagram of PI3K subunits and domains illustrating the regulatory p85 and catalytic p110 subunits. Inter-SH2 (iSH2) domain in p85 subunit interacts with p110. Binding of nerve growth factor (NGF) to TrkA receptor triggers the translocation of PI3K to the PM, phosphorylation of phosphoinositide 4,5-bisphosphate (PI(4,5)$P_2$) to PI(3,4,5)$P_3$, and fusion of TRPV1-containing vesicles with the PM. (**B, C**) Schematic diagram for OptoPI3K system using PIF-YPF or PIF-iSH2-YFP. PhyB-mCherry is tethered to the PM using CAAX lipidation (magenta star). The iSH2 domain of p85 is fused to PIF so that translocation of PIF-iSH2-YFP, together with endogenous p110, to the PM promotes PI(3,4,5)$P_3$ synthesis upon 650 nm light. (**D**) Monitoring PIF-iSH2-YFP translocation to and from the PM with 650 and 750 nm light, respectively (top, yellow). Synthesis of PI(3,4,5)$P_3$ follows PIF-iSH2-YFP translocation to the PM, as indicated by the localization of the PI(3,4,5)$P_3$ probe Akt-PH-CFP (bottom, sky blue). F-11 cells transiently expressing PhyB-mCherry-CAAX, PIF-iSH2-YFP, and Akt-PH-CFP were illuminated with 750 or 650 nm light as indicated with the upper bar. Collected traces of PIF-iSH2-YFP and Akt-PH-CFP normalized to the initial baselines during the first episode of 750 nm illumination. The black line indicates the mean of the data and the colored envelope represents the standard error of the mean (n=8). Because of the very low density of PI(3,4,5)$P_3$ present in the PM even in light- or NGF-stimulated cells (**Auger et al., 1989**), we used total internal reflection fluorescence (TIRF) microscopy to measure PI(3,4,5)$P_3$ density instead of confocal microscopy. TIRF illumination decreases exponentially with distance from the coverslip, selectively illuminating and exciting fluorophores within ~150 nm of the PM (**Lakowicz, 2006**; **Mattheyses and Axelrod, 2006**). (**E**) Scatter plot of PIF-iSH2-YFP and Akt-PH-CFP fluorescence for individual cells. Each point represents the 20 s average for 750 nm (2.66–3 min), 650 nm (5.66–6 min), and 750 nm (8.66–9 min). Translocation of both PIF-iSH2-YFP and Akt-PH-CFP is reversible.

The online version of this article includes the following source data and figure supplement(s) for figure 1:

**Source data 1.** Excel data for the time courses (**Figure 1D and E**).

**Figure supplement 1.** PIF-YFP translocates to plasma membrane (PM) in response to 650 nm light and back to the cytosol in response to 750 nm light.

**Figure supplement 1—source data 1.** Confocal images (.lsm) generated by Zeiss 710 confocal microscope (**Figure 1—figure supplement 1B**).

**Figure supplement 1—source data 2.** Excel data for the time course (**Figure 1—figure supplement 1C**).

**Figure supplement 2.** Detection of phosphoinositide 3,4,5-trisphosphate (PI(3,4,5)$P_3$) generated by PIF-iSH2-YFP at the plasma membrane (PM) using GRP1-PH-CFP.

**Figure supplement 2—source data 1.** Excel data for the time courses (**Figure 1—figure supplement 2A and B**).

## Approaches to studying membrane protein trafficking

The properties of conventional fluorescent labels have posed a significant barrier to understanding trafficking of membrane proteins, including TRPV1. Previous studies have used large tags to label the channel protein, typically a fluorescent protein (e.g. GFP) fused to the intracellular N- or C-terminus. Fusion to fluorescent proteins carries a number of disadvantages. Fluorescent proteins are relatively dim and are susceptible to bleaching. We and others have previously shown that the intracellular termini participate in critical protein-protein interactions, some of which are involved in channel trafficking (*Morenilla-Palao et al., 2004*; *Stein et al., 2006*; *Jeske et al., 2008*; *Camprubí-Robles et al., 2009*; *Jeske et al., 2009*; *Xing et al., 2012*; *Gregorio-Teruel et al., 2015*; *Mathivanan et al., 2016*; *Meng et al., 2016*). The presence of large fluorescent proteins could perturb these interactions (*Tsien, 1998*; *Montecinos-Franjola et al., 2020*). TRPV1 is expressed heavily in intracellular membranes, making it nearly impossible to optically isolate just those channels on the PM (*Liu et al., 2003*; *Gallego-Sandín et al., 2009*). Fluorescent anti-TRPV1 antibodies have been used to label extracellular epitopes to distinguish these populations (e.g. *Meng et al., 2016*; *Nakazawa et al., 2021*), but antibodies have the same problem of large size, and we have found that antibodies that recognize extracellular regions of TRPV1 are not very specific.

Click chemistry offers a rapid, specific, and flexible method for labeling of proteins in living cells (*Lang and Chin, 2014*; *Nikić et al., 2015*; *Liu and Kenry, 2019*). We have previously incorporated click chemistry-compatible noncanonical amino acids (ncAAs) into amino acid positions in the cytoplasm of mammalian cells and shown we can achieve highly efficient and rapid labeling when introducing the appropriate click chemistry probe into the medium (*Jang et al., 2020*; *Jana et al., 2023*). This approach requires co-expression of the target gene with an amber stop codon (TAG) with a plasmid encoding an evolved aminoacyl tRNA synthetase that is orthogonal to mammalian cells, together with its cognate tRNA (*Chin, 2014*; *Uttamapinant et al., 2015*). An additional plasmid encoding a dominant negative form of eukaryotic dominant negative elongation release factor can be expressed to increase efficiency of ncAA incorporation (*Schmied et al., 2014*). Incorporating a click chemistry ncAA into an extracellular position in a membrane protein and then applying a membrane-impermeant click chemistry-conjugated fluorophore would allow specific labeling of protein at the surface, even for those proteins that localize to both the surface and intracellular membranes (*Gregory et al., 2016*; *Mateos-Gil et al., 2016*; *Neubert et al., 2018*; *Ojima et al., 2021*). The recent development of click chemistry ncAAs and fluorescent probes that react very rapidly ($>10^4$ M$^{-1}$s$^{-1}$) make this an attractive approach for tracking membrane protein trafficking in real time (*Peng and Hang, 2016*; *Row and Prescher, 2018*; *Meineke et al., 2020*; *Jana et al., 2023*).

In this work, we leverage optogenetics to activate PI3K with light, genetic code expansion to incorporate a click chemistry ncAA, and a new, membrane-impermeant click chemistry-conjugated fluorophore to interrogate the mechanism by which NGF induces trafficking of TRPV1 to the PM. By isolating the PI3K pathway downstream of NGF from the other pathways coupled to TrkA (PLCγ and MAP/ERK), we demonstrate that PI3K activity is sufficient to increase TRPV1 trafficking to the PM. We apply the same approach to InsR, to demonstrate that our new approach is of general use for different types of membrane proteins, and show that activation of PI3K is sufficient to increase InsR trafficking to the PM. Given the importance of InsR recycling in the development of insulin resistance, understanding the role of PI3K in regulating InsR expression on the PM is critical.

## Results

Here, we used the previously published phytochrome B (PhyB)/phytochrome interacting factor (PIF) system to activate and deactivate PI3K with light (*Figure 1B*; *Levskaya et al., 2009*; *Toettcher et al., 2011*). Uncoupling PI3K from activation of TrkA allowed us to study the effects of PI3K on membrane protein trafficking without activating other signaling pathways downstream of TrkA. To apply the PhyB/PIF system to control PI3K activity, we fused the iSH2 domain of the p85 regulatory subunit of PI3K to PIF (*Figure 1C*). It has previously been shown that the iSH2 domain associates with the endogenous p110 catalytic subunit of PI3K (*Klippel et al., 1993*). Because this heterodimer is missing the autoinhibitory domains of p85, it is constitutively active, so that PI3K phosphorylates its substrate, PI(4,5)P$_2$ to make PI(3,4,5)P$_3$ whenever the PIF-iSH2-YFP is driven to the membrane (*Figure 1C*; *Suh et al., 2006*; *Idevall-Hagren et al., 2012*). The iSH2 domain has been previously expressed as a fusion

protein with PIF and has been shown to reversibly translocate to the PM and activate PI3K in response to light (*Figure 1—figure supplement 1*; *Levskaya et al., 2009*; *Toettcher et al., 2011*).

Using a cyan fluorescent protein (CFP)-labeled pleckstrin homology domain from the enzyme Akt (Akt-PH-CFP), which specifically binds the PI(3,4,5)$P_3$ product of PI3K (*Lemmon, 2008*), as a probe for PI(3,4,5)$P_3$, we confirmed that the PhyB/PIF-iSH2 machinery could be used to rapidly generate PI(3,4,5)$P_3$ in response to 650 nm light (*Figure 1D and E*). In response to 750 nm light, PIF-iSH2-YFP was released from the membrane and the levels of PI(3,4,5)$P_3$ returned to baseline, presumably due to the activity of the endogenous PI(3,4,5)$P_3$ phosphatase PTEN. Similar experiment using GRP1-PH-CFP, another PI(3,4,5)$P_3$ probe yielded the same results (*Figure 1—figure supplement 2*).

## Activation of PI3K, without other pathways downstream of NGF/TrkA, is sufficient to drive trafficking of TRPV1 to the PM

We and others have previously demonstrated that activation of the RTK TrkA by NGF leads to increased trafficking of the ion channel TRPV1 to the PM (*Zhang et al., 2005*; *Stein et al., 2006*; *Zhu and Oxford, 2007*; *Stratiievska et al., 2018*) This increase in surface expression is associated with increased sensitivity to painful stimuli sensed by TRPV1. As shown in *Figure 2A and B*, the NGF-induced trafficking of TRPV1 follows the increase in PM PI(3,4,5)$P_3$ levels. Although TrkA couples with PLCγ and the MAP/ERK pathways in addition to PI3K, significant evidence indicates that PI3K activation is *necessary* to NGF-induced trafficking of TRPV1 to the PM (*Zhang et al., 2005*; *Stein et al., 2006*; *Zhu and Oxford, 2007*; *Stratiievska et al., 2018*).

We used PhyB with PIF-iSH2 to test the hypothesis that PI3K activity is *sufficient* to drive trafficking of TRPV1 to the PM. Together with PhyB, PIF-iSH2 (without a fluorescent tag), and Akt-PH-CFP, we expressed TRPV1 fused to yellow fluorescent protein (YFP). Stimulation of cells with 650 nm light gave the expected rise in PM-associated Akt-PH-CFP fluorescence and, importantly, a rise in PM-associated TRPV1-YFP fluorescence (*Figure 2C and D*). The light-induced rise in PM PI(3,4,5)$P_3$ and TRPV1 qualitatively resembled those induced by NGF. Because PLCγ and the MAP/ERK were not directly activated by the 650 nm light, we conclude that activation of the PI3K pathway is sufficient to cause TRPV1 trafficking to the PM.

## TRPV1 prevents dissociation of PIF-iSH2-YFP from the membrane in response to 750 nm light

In monitoring localization of PIF-iSH2-YFP in light-activated PI3K experiments in TRPV1-expressing cells, we found that PIF-iSH2-YFP no longer translocated to the cytoplasm in response to 750 nm light (*Figure 2—figure supplement 1A and B*). Indeed, the small amount of mCherry excitation light (561 nm) used to identify PhyB-mCherry-positive cells seemed to be sufficient to cause a slow accumulation of PIF-iSH2-YFP at the PM even during continuous exposure of the cell to 750 nm light (*Figure 2—figure supplement 1C and D*). Not surprisingly, with PIF-iSH2-YFP accumulating at the PM during imaging of TRPV1-expressing cells, the 650 nm light-induced increase in PM-associated Akt-PH-CFP did not reverse upon exposure to 750 nm light (*Figure 2C and D*). TRPV1 expression had no effect on a version of PIF-YFP that lacked the iSH2 domain (*Figure 2—figure supplement 2*), indicating that irreversible localization to the PM required both the iSH2 fragment of PI3K and TRPV1. Expression of the related TRPM4 ion channels, which do not include an ARD, also had no effect on 750 nm light-induced dissociation of PIF-iSH2-YFP from the PM (*Figure 2—figure supplement 3*), indicating that this effect is specific to TRPV1. We have previously shown that the ARD of TRPV1 interacts directly with the p85 subunit of PI3K from which iSH2 is derived (*Stein et al., 2006*). We speculate that the PIF-iSH2-YFP in TRPV1-expressing cells might not dissociate from the PM when stimulated with 750 nm light because it encounters TRPV1 at the PM and binds to it, remaining bound to TRPV1 even after releasing PhyB.

## Resolving changes in surface expression requires improved approaches

Total internal fluorescence (TIRF) microscopy was needed to resolve the NGF- and light-induced increase in PM PI(3,4,5)$P_3$ levels but is a poor approach for distinguishing membrane protein localization in the PM from that in intracellular compartments (*Stratiievska et al., 2018*). For membrane proteins with very high endoplasmic reticulum localization, like TRPV1 (*Tominaga et al., 1998*),

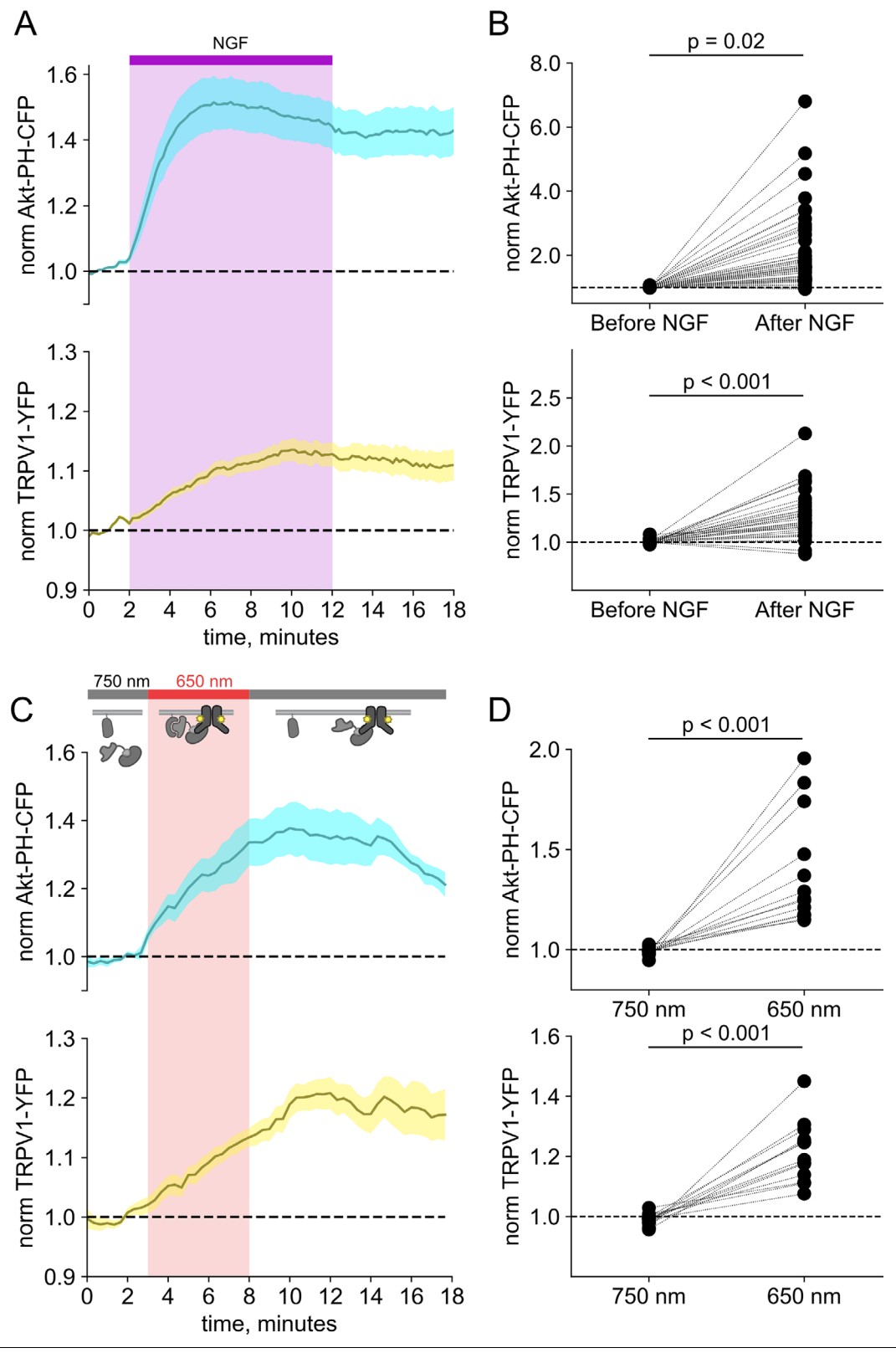

**Figure 2.** Activation of phosphoinositide 3-kinases (PI3K) with light is sufficient to induce trafficking of TRPV1 to the plasma membrane (PM). Simultaneous total internal fluorescence (TIRF) measurement of phosphoinositide 3,4,5-trisphosphate (PI(3,4,5)P₃) (cyan) and TRPV1 (yellow) in the PM in response to either (**A**) nerve growth factor (NGF) or (**C**) light. (**A**) F-11 cells were transfected with TrkA/p75NTR, Akt-PH-CFP, and TRPV1-YFP. NGF (100 ng/mL)

*Figure 2 continued on next page*

*Figure 2 continued*

was applied during the times indicated by the bar/shading. Plotted are the PM-associated fluorescence in Akt-PH-CFP (top, cyan) and TRPV1-YFP (bottom, yellow) within the cell footprints. Data are reproduced from *Stratiievska et al., 2018*. (**C**) F-11 cells transfected with PhyB-mCherry-CAAX, PIF-iSH2 (without a fluorescent tag), Akt-PH-CFP, and TRPV1-YFP were illuminated with 750 or 650 nm light as indicated. Color scheme as in (**A**), with line indicating the mean and envelope indicating the standard error of the mean (n=13 for Akt-PH-CFP; n=16 for TRPV1-YFP). Note the poor or irreversible increase of PM PI(3,4,5)P$_3$ in the PM. Inset cartoons depict the model for retention of iSH2 at the PM via binding to TRPV1. (**B, D**) Scatter plots for individual cells. Each point represents the 20 s average for before (0.83–1.17 min) and after NGF (14.8–15.2 min) or for before (1.31–1.63 min) and after 650 nm (10.8–11.2 min).

The online version of this article includes the following source data and figure supplement(s) for figure 2:

**Source data 1.** These data are reproduced from *Stratiievska et al., 2018* (*Figure 2A and B*).

**Source data 2.** Excel data for time courses and scatter plots (*Figure 2C and D*).

**Figure supplement 1.** 750 nm light fails to cause inter-SH2 (iSH2) dissociation from the plasma membrane (PM) in TRPV1-expressing cells.

**Figure supplement 1—source data 1.** Excel data for time courses and scatter plots in F-11 and HEK cells.

**Figure supplement 2.** PIF-YFP dissociates from the plasma membrane (PM) in response to 750 nm light even in TRPV1-expressing cells.

**Figure supplement 2—source data 1.** Excel data for time course and scatter plot.

**Figure supplement 3.** 750 nm light succeeds in causing inter-SH2 (iSH2) dissociation from the plasma membrane (PM) in TRPM4-expressing cells.

**Figure supplement 3—source data 1.** Excel data for time courses and scatter plots in F-11 and HEK cells.

this optical contamination leads to an underestimate of true changes in surface localization in TIRF experiments.

To circumvent contamination of TIRF signals on the PM with fluorescence from intracellular compartments, we implemented an inverse-electron-demand Diels-Alder cycloaddition click chemistry approach to selectively label PM localized proteins (*Arsić et al., 2022*). For this application, we used genetic code expansion to incorporate an ncAA at an extracellular site (*Neubert et al., 2018*; *Bessa-Neto et al., 2021*; *Kuhlemann et al., 2021*). We used Tet3-Bu (*Figure 3A*), a tetrazine-containing amino acid which can be efficiently labeled with a cyclopropane-fused strained trans-cyclooctene (sTCO) with no detectable off-target reactivity within the time scale of our experiments (see below) (*Jang et al., 2020*; *Jana et al., 2023*). Our goal was to express membrane proteins incorporating Tet3-Bu at an extracellular site and measure labeling with a membrane-impermeant sTCO-coupled dye (*Figure 3B*) under control conditions and then again after activation of PI3K with either NGF or 650 nm light.

We tested our click chemistry/ncAA approach using TRPV1 and the InsR, a model RTK we and others have previously used in studies with different ncAAs (*Nikić et al., 2015*; *Jones et al., 2021*). For both proteins, we used amber codon suppression to site-specifically incorporate Tet3-Bu (*Jang et al., 2020*) into an extracellular loop (T468Tet3-Bu for TRPV1 and K676Tet3-Bu for InsR). This involved introducing a TAG stop codon at the selected site within the coding sequence that would place the ncAA on the extracellular side of the PM. We then co-expressed one plasmid encoding either TRPV1-T468TAG or InsR-K676TAG fused to GFP, a second plasmid encoding an aminoacyl tRNA synthetase evolved to incorporate the ncAA and the orthogonal tRNA, and a third plasmid encoding a dominant negative elongation release factor, which increased the efficiency of Tet3-Bu incorporation (*Schmied et al., 2014*). The cell culture medium was then supplemented with Tet3-Bu. As indicated by the GFP signal and in-gel fluorescence both TRPV1-T468Tet3-Bu-GFP and InsR-K676Tet3-Bu-GFP expressed at high levels only in transfected cells (*Figure 3C and E* and *Figure 3—figure supplement 1C*). To test whether incorporation of Tet3-Bu interfered with the function of TRPV1, we measured capsaicin-activated currents with whole-cell voltage clamp and found that TRPV1-T468Tet3-Bu-GFP gave robust capsaicin-activated currents (*Figure 3—figure supplement 1A and B*). To determine whether InsR-K676TAG protein retained its function, we measured insulin-induced endocytosis (*Carpentier et al., 1992*) and found endocytosis in response to insulin was maintained (*Figure 3—figure supplement 2*) with approximately the expected kinetics (*Figure 3—video 1*).

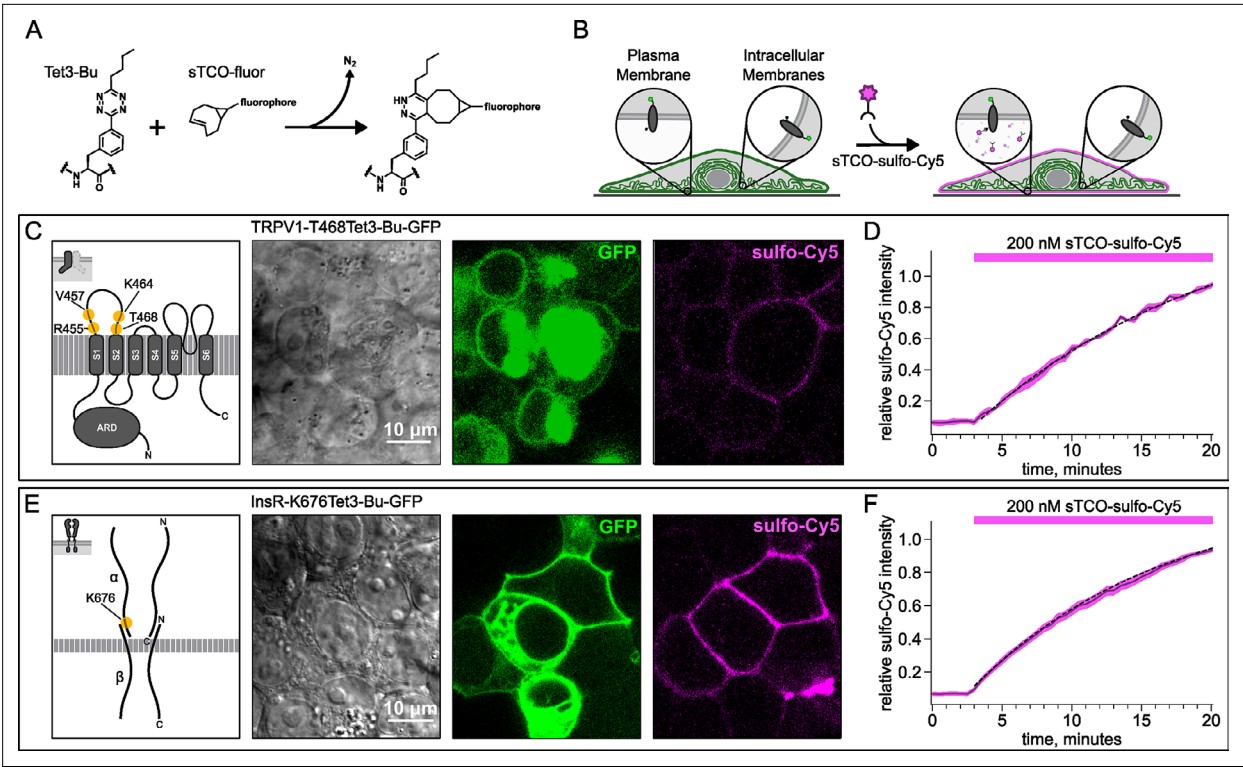

**Figure 3.** Labeling the TRPV1 and InsR with membrane-impermeant sTCO-Cy5. Confocal imaging illustrates the labeling of membrane proteins incorporating the noncanonical amino acid (ncAA) Tet3-Bu with sTCO-sulfo-Cy5 in HEK293T/17 cells. The membrane-impermeable dye labeled only the proteins on the plasma membrane (PM). (**A**) Schematic of the reaction between Tet3-Bu and sTCO-conjugated dyes. (**B**) Cartoon representing the selective labeling of membrane proteins incorporating Tet3-Bu at an extracellular site with membrane-impermeant sTCO-sulfo-Cy5. (**C, E**) Confocal images of HEK293T/17 cells expressing (**C**) TRPV1-468Tet3-Bu-GFP or (**E**) InsR-676Tet3-Bu-GFP. GFP fluorescence reflects expression of the proteins in the confocal volume across the field of view. Initially the cells did not show any detectable Cy5 fluorescence but after incubation of several minutes of 200 nM sTCO-sulfo-Cy5 showed Cy5 fluorescence at the PM. The Cy5 images shown for (**C**) TRPV1-Tet3-Bu and (**E**) InsR-Tet3-Bu were obtained at the end of the experiment (20 min). (**D, F**) The graphs summarize the Cy5 fluorescence at the PM in (**D**) TRPV1-Tet3-Bu-GFP or (**F**) InsR-Tet3-Bu-GFP-expressing cells. Solid traces represent the mean and envelopes the standard error of the mean (n=3 for TRPV1 and n=11 for InsR). Dashed traces represent a fit to the mean with a single exponential (tau = 17.8 min for TRPV1; tau = 13.8 min for InsR). Fits to the individual time courses for all the cells gave a mean of 18.2 min for TRPV1 (±2.2 min) and 19.3 min for InsR (±4.0 min).

The online version of this article includes the following video, source data, and figure supplement(s) for figure 3:

**Source data 1.** Original confocal microscopic images for TRPV1 labeling (*Figure 3C*).

**Source data 2.** Original confocal microscopic images for InsR labeling (*Figure 3E*).

**Source data 3.** Excel data for scatter plots for TRPV1 and InsR labeling (*Figure 3D and F*).

**Figure supplement 1.** Incorporation of Tet3-Bu into TRPV1 and InsR.

**Figure supplement 1—source data 1.** Representative whole-cell patch clamp recordings.

**Figure supplement 1—source data 2.** Whole-cell electrophysiology data from all cells with peak capsaicin-activated currents.

**Figure supplement 1—source data 3.** Original gel image with Coomassie blue staining, original fluorescent gel image, and labeled gel images.

**Figure supplement 2.** Activity of InsR-K676TAG measure as its recycling upon insulin treatment.

**Figure supplement 2—source data 1.** Original confocal microscopic images for InsR endocytosis before (9th image) and after insulin (22nd image) treatment.

**Figure supplement 2—source data 2.** Excel data for scatter plot for InsR endocytosis (*Figure 3—figure supplement 2B*).

**Figure supplement 3.** Labeling of cells with sTCO conjugates of TAMRA, JF 646, fluorescein, and sulfo-Cy5.

**Figure supplement 3—source data 1.** Original confocal microscopic images for TAMRA (90th image, plus GFP and bright field), JF 646 (72nd image, plus GFP and bright field), fluorescein (126th image, plus GFP and bright field), and Cy5 (115th image, plus GFP and bright field).

**Figure supplement 3—source data 2.** Excel data for labeling kinetics of sTCO dyes (*Figure 3—figure supplement 3A–D*).

**Figure 3—video 1.** Activity of InsR-K676TAG measure as its recycling upon insulin treatment.

https://elifesciences.org/articles/91012/figures#fig3video1

## Developing a membrane-impermeant sTCO dye

sTCO bearing functional groups such as spin labels and fluorophores have been used previously for in-cell studies (*Murrey et al., 2015*; *Ryan et al., 2022*; *Jana et al., 2023*). To determine whether existing sTCO-conjugated dyes would give the required PM-selective labeling, we tested whether sTCO-TAMRA (*Jang et al., 2020*) or sTCO-JF646 (*Jana et al., 2023*) were membrane impermeant. We also synthesized and tested sTCO-fluorescein. Unfortunately, these compounds readily equilibrated across the cell PM at the concentrations required for labeling, giving indistinguishable cytosolic fluorescence in untransfected and transfected cells (*Figure 3—figure supplement 3A–C*). We therefore synthesized a new sTCO-sulfo-Cy5 conjugate, which contains two negative charges at physiological pH to minimize passive diffusion through membrane and therefore selectively label membrane proteins at the cell surface (*Figure 3—figure supplement 3D*; *Hoffmann et al., 2015*; *Nikić et al., 2015*; *Kozma et al., 2016*; *Lam et al., 2018*; *Keller et al., 2020*).

To determine whether sTCO-sulfo-Cy5 would selectively label a membrane protein with an extracellular Tet3-Bu, we measured Cy5 fluorescence in the presence of 200 nM sTCO-sulfo-Cy5 in the bath, a low concentration that gives very low background fluorescence. The click chemistry reaction between Tet3-Bu incorporated at the extracellular sites and sTCO-sulfo-Cy5 effectively concentrated the dye at the PM, allowing significant membrane labeling to be observed for both membrane proteins examined (TRPV1-T468Tet3-Bu-GFP – *Figure 3C* and InsR-K676Tet3-Bu-GFP – *Figure 3E*). Importantly, sTCO-sulfo-Cy5 did not appear to equilibrate across the cell membrane and did not label untransfected cells (i.e. those without GFP; *Figure 3—figure supplement 3D*). In vitro, the click chemistry reaction between free Tet3-Bu ncAA and sTCO occurs with a rate of $2 \times 10^4$ $M^{-1}s^{-1}$; this reaction rate increased by fourfold when Tet3-Bu was incorporated on the surface of GFP (*Jang et al., 2020*).

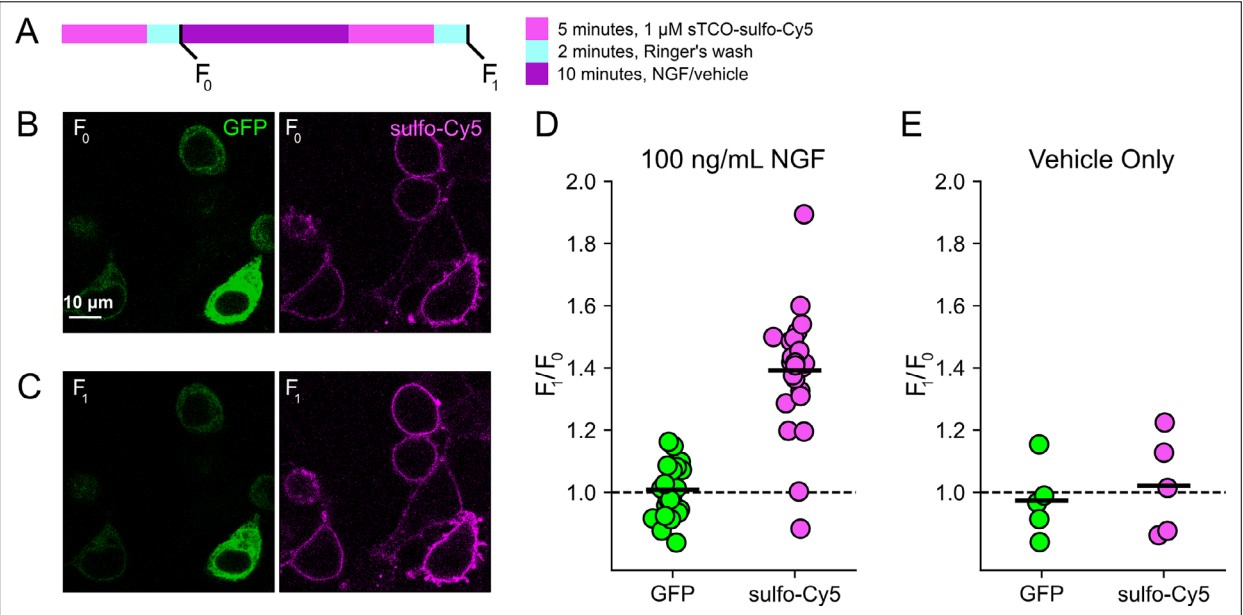

**Figure 4.** Click chemistry labeling of TRPV1-468Tet3-Bu-GFP with sTCO-Cy5 to measure nerve growth factor (NGF)-induced trafficking of TRPV1 to the plasma membrane (PM). HEK293T/17 cells expressing TRPV1-Tet3-Bu-GFP and NGF receptor were labeled with extracellular sTCO-sulfo-Cy5 and inspected with confocal microscopy. (**A**) Experimental protocol. Cells were incubated with 1 μM sTCO-sulfo-Cy5 for 5 min and free dye removed from the bath by washing for 2 min with dye-free Ringer's solution ('pulse-chase' labeling, $F_0$). Then the cells were treated with 100 ng/mL NGF for 10 min before the second sulfo-Cy5 labeling ($F_1$). Confocal images after initial sTCO-sulfo-Cy5 labeling (**B**) and after the 10 min treatment with NGF and subsequent sTCO-sulfo-Cy5 labeling (**C**). (**D**) Summary scatter plot from multiple measurements, with individual experiments shown as dots and the mean of the experiments as black bars. The effect of NGF on GFP and sulfo-Cy5 signals is presented as a ratio of $F_1/F_0$ (n=24). After NGF treatment, the ratio increased for sulfo-Cy5 significantly (p<0.001) but not for GFP (p=0.64). (**E**) The same experiment without NGF treatment ('vehicle only', n=5). Vehicle treatment did not change both GFP (p=0.63) and sulfo-Cy5 (p=0.78).

The online version of this article includes the following source data for figure 4:

**Source data 1.** Original confocal microscopic images for TRPV1 trafficking.

**Source data 2.** Excel data for scatter plot for TRPV1 trafficking (*Figure 4D and E*).

The low background fluorescence recorded with 200 nM dye in the bath allowed us to measure the rate of labeling for TRPV1-468Tet3-Bu-GFP and InsR-646Tet3-Bu-GFP with sTCO-sulfo-Cy5. As shown in *Figure 3D–F*, the reaction rate we measured was approximately $5 \times 10^3$ $M^{-1}s^{-1}$ for both proteins, somewhat slower than expected for a protein in solution and on par with the reaction between the free amino acid and sTCO in solution (*Meijer et al., 1998*; *Taylor et al., 2011*).

## Click chemistry labeling can be used to measure NGF-induced trafficking of TRPV1 to the PM

Based on the extensive contamination of TIRF signals with fluorescence from the ER and other intracellular compartments (*Stratiievska et al., 2018*), we hypothesized that our previous measurements of NGF-induced trafficking of TRPV1 to the PM using TIRF represent underestimates of the true changes in PM TRPV1. If this were the case, using click chemistry to resolve NGF-induced changes in PM TRPV1 should reveal a greater increase in PM TRPV1. At the beginning of the experiment, we labeled surface TRPV1-468Tet3-Bu-GFP with a pulse of 1 µM sTCO-sulfo-Cy5 and then washed the label from the bath (*Figure 4A*). We used this higher concentration of sTCO dye (five times higher than in *Figure 3*) to label the PM channels more rapidly, as the NGF-induced trafficking of TRPV1 to the PM occurs with kinetics comparable to labeling with 200 nM sTCO-sulfo-Cy5 (compare *Figure 2A* to *Figure 3D*). We then treated the cells with NGF for 10 min and again pulse labeled the surface channels with a brief exposure to 1 µM sTCO-sulfo-Cy5. We observed an ~1.4-fold increase in PM Cy5 staining, indicating that NGF induced trafficking of TRPV1 to the surface (*Figure 4B–D*). In contrast, the total number of TRPV1 channels did not change, as evidenced by the total intensity of GFP fluorescence, which was not affected by NGF. This NGF-induced increase in surface expression is indeed greater than the ~1.1-fold increase in NGF-induced surface expression of TRPV1 observed in TIRF experiments (*Figure 2A and B*). In control experiments in which cells were exposed to the vehicle (without NGF), neither the GFP signal nor the sTCO-sulfo-Cy5 signal changed (*Figure 4E*), indicating that the NGF-induced increase in sTCO-sulfo-Cy5 was specific to activation of the TrkA receptor by NGF.

## Click chemistry labeling can be used to measure light-induced trafficking to the PM

We next asked whether click chemistry labeling could improve the resolution of PM trafficking in a system in which we have decoupled PI3K activity from other pathways downstream of NGF, i.e., the PhyB/PIF system. Results from these experiments with TRPV1-468Tet3-Bu-GFP and InsR-676Tet3-Bu-GFP are shown in *Figure 5*. After first labeling cells with sTCO-sulfo-Cy5, we exposed cells to an initial 10 min exposure to 750 nm light and labeled any newly delivered protein at the PM using a second treatment with sTCO-sulfo-Cy5 (*Figure 5A*). The GFP fluorescence remained the same under all conditions indicating that total protein expression did not change over the course of the experiments. For TRPV1-T468Tet3-Bu-GFP, the sTCO-sulfo-Cy5 labeling increased slightly in response to the initial exposure to 750 nm light (*Figure 5B and D*). This small increase in sTCO-sulfo-Cy5 labeling was specific to TRPV1-T468Tet3-Bu, as 750 nm light produced no change in sTCO-sulfo-Cy5 labeling of InsR-K676Tet3-Bu-GFP (*Figure 5F and H*). The TRPV1-specific increase in labeling after 750 nm illumination is likely due to the small, irreversible PM localization of PIF-iSH2-YFP that occurs in TRPV1-expressing cells (*Figure 2—figure supplement 1*). This small increase in sTCO-sulfo-Cy5 labeling of TRPV1-T468Tet3-Bu was also seen in control experiments using two sequential exposures to 750 nm light (*Figure 5—figure supplement 1*). Note that the increase of TRPV1-T468Tet3-Bu labeling, $F_2/F_1$, in experiments with 650 nm light (*Figure 5*) was significantly greater than in experiments with only 750 nm light (p<0.001; *Figure 5—figure supplement 1*).

After the initial control period with 750 nm light, we exposed the cells to 650 nm light for 10 min to activate PI3K and then labeled newly delivered protein at the PM with sTCO-sulfo-Cy5. As shown in *Figure 5B–E* for TRPV1-T468Tet3-Bu and *Figure 5F–I* for InsR-K676Tet3-Bu-GFP, activation of PI3K with 650 nm light increased expression of protein on the PM. For TRPV1, the amplitude of the increase in surface expression, ~1.4-fold, was similar to that observed in click chemistry experiments in response to NGF (*Figure 4*). Surprisingly, the increase in surface expression of InsR-K676-Tet3-Bu was also about 1.4-fold. Like the NGF receptor TrkA, InsR is an RTK whose activation stimulates PI3K activity (*Saltiel, 2021*). Binding of insulin to InsR is a well-studied signal for InsR endocytosis (*Carpentier et al., 1992*), and PI3K activation downstream of InsR is key to delivery of GLUT4 transporters to

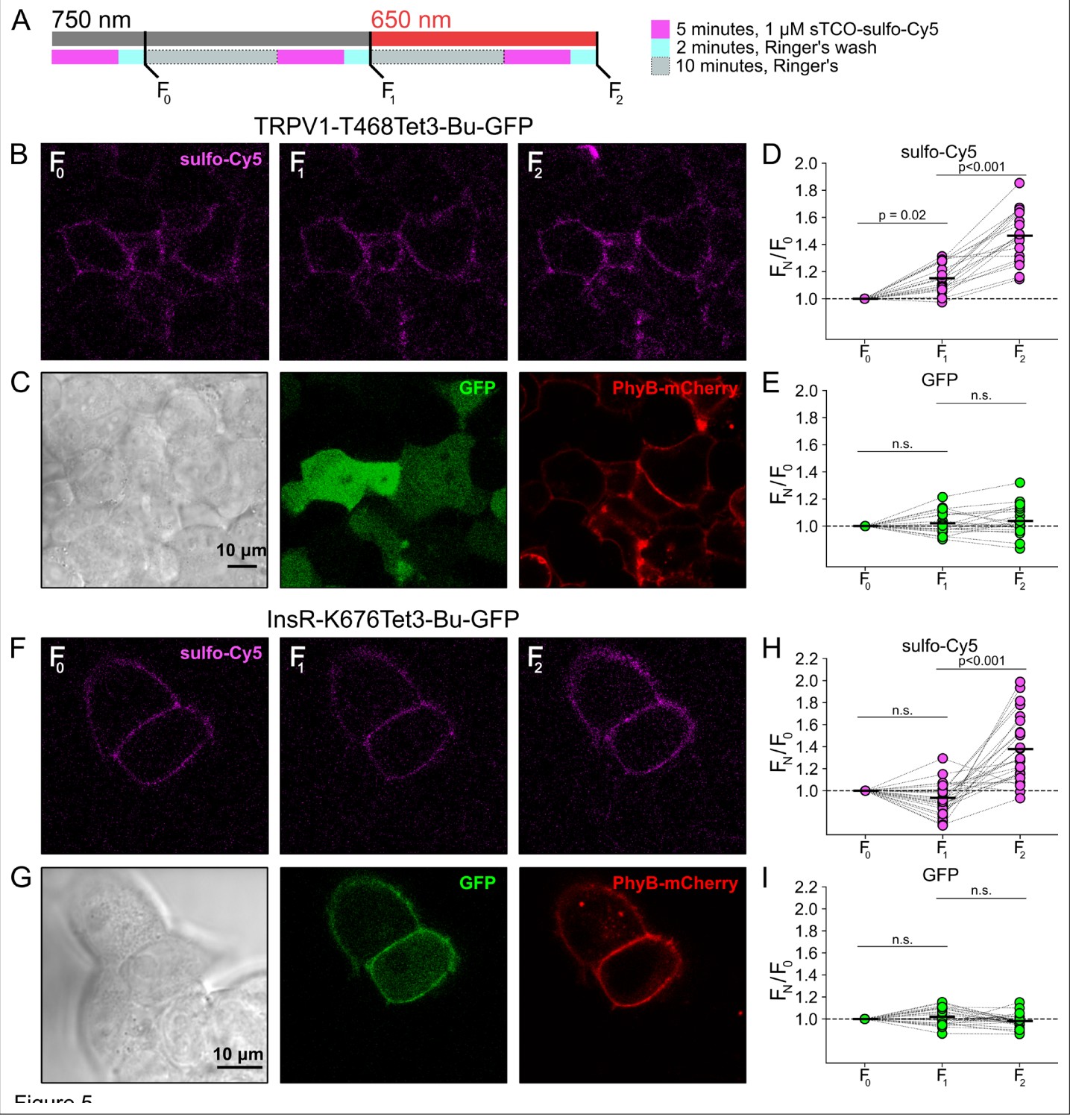

**Figure 5.** Measuring light-activated phosphoinositide 3-kinases (PI3K)-induced TRPV1 and InsR trafficking to the plasma membrane (PM) using click chemistry. (**A**) Illustration of the experimental protocol. HEK293T/17 cells expressing TRPV1-468Tet3-Bu-GFP and InsR-676Tet3-Bu-GFP. (**B**) Confocal images of sulfo-Cy5 obtained at different stages as depicted in (**A**). (**C**) Bright-field (left) and confocal (middle and right) images obtained at the end of experiment. Comparison of bright field (left) and GFP (middle) distinguishes TRPV1-Tet3-Bu-GFP-expressing cells from untransfected cells. PhyB-mCherry images (red) indicate that most TRPV1-positive cells expressed significant levels of the PhyB/PIF machinery for activating PI3K. (**D, E**) Summary scatter plots from multiple experiments, with individual cells shown as dots and the mean shows as black lines (n=20). The effects of 750 and 650 nm illumination on sulfo-Cy5 (**D**) and GFP (**E**) are presented as ratios of fluorescence intensity after illumination with the indicated wavelength of illumination to the initial fluorescence. (**F–I**) The same experiment for InsR-676Tet3-Bu-GFP trafficking (n=24).

*Figure 5 continued on next page*

*Figure 5 continued*

The online version of this article includes the following source data and figure supplement(s) for figure 5:

**Source data 1.** Original confocal microscopic images for TRPV1 trafficking (*Figure 5B and C*).

**Source data 2.** Scatter plot for TRPV1 trafficking (*Figure 5D and E*).

**Source data 3.** Original confocal microscopic images for InsR trafficking (*Figure 5F and G*).

**Source data 4.** Excel data for scatter plot for InsR trafficking (*Figure 5H and I*).

**Figure supplement 1.** Control experiment for light-activated phosphoinositide 3-kinases (PI3K)-induced TRPV1 and InsR trafficking to the plasma membrane (PM).

**Figure supplement 1—source data 1.** Original confocal microscopic images for TRPV1 trafficking (*Figure 5—figure supplement 1B and C*).

**Figure supplement 1—source data 2.** Excel data for scatter plot for TRPV1 trafficking (*Figure 5—figure supplement 1D and E*).

**Figure supplement 1—source data 3.** Original confocal microscopic images for InsR trafficking (*Figure 5—figure supplement 1F and G*).

**Figure supplement 1—source data 4.** Excel data for scatter plot for InsR trafficking (*Figure 5—figure supplement 1H and I*).

the PM (*Saltiel, 2021*). However, to our knowledge, PI3K activity has not been previously known to stimulate InsR delivery to the PM. Although it would be exceedingly interesting if insulin binding to InsR could trigger both InsR endocytosis *and* exocytosis, further studies will be required to determine the physiological significance, if any, of PI3K-stimluated delivery of InsR to the PM.

## Discussion

Here, we present proof-of-principle experiments demonstrating the simultaneous use of an optogenetic approach for PI3K activation and genetic code expansion/click chemistry to interrogate regulation of membrane protein trafficking. This combination of approaches is especially useful for membrane proteins with significant expression in the endoplasmic reticulum and/or other intracellular compartments, which makes it very difficult to identify proteins in the PM definitively when using fusions with GFP or similar genetically encoded labels. In addition to distinguishing proteins at the surface, using a small ncAA and click chemistry is likely much less perturbing to protein structure and function than fusion with an ~20–30 kDa protein such as GFP, SNAP-tag, or HaloTag. Restricting the ncAA and fluorescent label to an extracellular region/loop is also less likely to interfere with protein-protein interactions on the intracellular side, compared with a large protein fusion to an intracellular N- or C-terminal of a protein of interest.

Successful implementation of this approach required overcoming a number of barriers: efficient incorporation of Tet3-Bu into the extracellular sites; demonstration that the Tet3-Bu-incorporating protein had the expected functional properties; development of a membrane-impermeant sTCO-coupled dye; and optimization of the PhyB/PIF and genetic code expansion machinery in the same cells. Incorporation of Tet3-Bu at the extracellular side of TRPV1 was well tolerated in the S1-S2 loop, with two of four sites yielding functional, capsaicin-activated channels. We have found incorporation efficiency to be difficult to predict. Screening through desirable sites typically is the best approach, with approximately one half to one third of selected sites showing reasonable levels of ncAA incorporation.

There is significant room for improvement of these methods, particularly in developing more rapid labeling of the ncAA. As shown in *Figure 3*, complete labeling with a low concentration of sTCO dye required more than 10 min. Monitoring membrane protein trafficking to the surface in real time will require a reaction that is at least 5- to 10-fold faster. Increasing the speed of labeling can be accomplished using $\geq 1$ μM sTCO dye, but at such high concentrations of dye the background signal from dye in solution interferes with imaging. We have recently developed new ncAAs with the tetrazine ring more proximal to the amino acid beta carbon, which we term Tet4, that react with sTCO-based labels with rates as fast as $10^6$ M$^{-1}$s$^{-1}$ (*Jana et al., 2023*). The aminoacyl tRNA synthetases that incorporate Tet4 amino acids into mammalian cells are not as efficient as those for Tet3 amino acids. Improving the efficiency of incorporating Tet4 amino acids would allow us to express Tet4-incorporating proteins and measure delivery of proteins to the PM in real time.

Another limitation to our approach is the need to co-express five to six plasmids to use PhyB/PIF and Tet/sTCO machinery simultaneously. Developing stable cell lines for ncAA incorporation would reduce

the need for multiple plasmid transfection and, perhaps, lead to more uniform expression across cell populations. The PhyB/PIF machinery requires supplementing cell medium with the phycocyanobilin (PCB) chromophore, making it incompatible with in vivo experiments. Introducing four genes, PcyA, H01, Fd, and Fnr, and knocking down/out biliverdin reductase A leads to efficient synthesis of PCB in mammalian cells (*Uda et al., 2017*; *Uda et al., 2020*). It is therefore possible that genetically modified animals might one day express the PhyB/PIF machinery and chromophore and allow optogenetic activation of PI3K in vivo. However, there is as yet no method for synthesizing the necessary ncAAs within cells, precluding in vivo use of the PhyB/PIF with tetrazine-based genetic code expansion.

PI3K is a universal signal for trafficking to the PM. In *Dictyostelium*, PI3K activation at the leading edge of cells underlies chemotaxis toward nutrients (*Nichols et al., 2015*). In macrophages, PI3K at the leading edge of cells drives motility toward the object of phagocytosis (*Hawkins et al., 2006*; *Hawkins and Stephens, 2015*). In the cases discussed here, membrane proteins constitute the cargo delivered to the PM in response to PI3K signaling. A vast literature has examined the mechanisms by which InsR is recycled from the PM in response to insulin (*Knutson, 1991*; *Ye et al., 2017*; *Chen et al., 2019*), but less is known about ways in which delivery of InsR to the PM is regulated. We show here, for the first time, that activation of PI3K is sufficient to increase InsR in the PM. Future studies will be needed to understand the role for this process in liver and muscle cells. For TRPV1, PI3K was identified as an important signal that enhances surface expression in response to NGF, but the steps between PI3K activation and fusion of vesicles containing TRPV1 cargo with the PM are a mystery. The tools developed here provide a means by which the cellular mechanisms underlying PI3K-mediated delivery of membrane proteins can be interrogated.

## Materials and methods
### Molecular biology and constructs used
The cDNAs used in this study were provided by: rat TRPV1 in pcDNA3.1 – David Julius, UCSF, San Francisco, CA, USA (*Caterina et al., 1997*) the human insulin receptor (accession AAA59452.1) with the K676TAG mutation and C-terminal GFP in a plasmid based on pEGFP – Edward Lemke, European Molecular Biology Laboratory (EMBL), Heidelberg, Germany (*Nikić et al., 2015*) eukaryotic elongation release factor 1 with E55D mutation in pCDNA5-FRT – Jason Chin, Medical Research Council Laboratory of Molecular Biology, Cambridge, England (*Schmied et al., 2014*) TrkA (rat) in the pcCMV5 vector and p75NTR (rat) in the pcDNA3 vector from Mark Bothwell, University of Washington, Seattle, WA, USA; PH-Akt-cCerulean in pcDNA3-k vector – Orion Weiner, UCSF, San Francisco, CA, USA (*Toettcher et al., 2011*) and *Methanosarcina barkeri* R2-84-RS aminoacyl tRNA synthetase/tRNA$_{CUA}$ in the pAcBac1 plasmid – Ryan Mehl, University of Oregon, Corvallis, OR, USA (*Jang et al., 2020*). Gibson cloning was used to generate the C-terminal GFP fusion for TRPV1 and to introduce 468TAG stop codon. All constructs were verified using Sanger sequencing.

In selecting the extracellular site for ncAA incorporation into TRPV1, we focused on the S1-S2 loop (*Figure 3C*) because the S3-S4 loop is very short (five amino acids) and we wished to avoid the pore turret between S5 and S6, which has been implicated in gating of the pore (*Yang et al., 2015*; *Bae et al., 2016*; *Jara-Oseguera et al., 2016*; *Ma et al., 2016*). In our initial screen, we tested incorporation at four positions, R455, V457, K464, and T468 (*Figure 3C*), and used a C-terminal GFP fusion to facilitate screening. We found that TRPV1-K464Tet3-Bu-GFP and TRPV1-T468Tet3-Bu-GFP produced robust, capsaicin-activated currents when interrogated with whole-cell patch clamp (*Figure 3—figure supplement 1A and B*), whereas TRPV1-R455Tet3-Bu-GFP and TRPV1-V457Tet3-Bu-GFP of incorporation did not (data not shown). We focused on TRPV1-T468Tet3-Bu-GFP, as TRPV1-K464Tat3-Bu-GFP was not efficiently labeled in subsequent experiments (data not shown). We next tested whether expression of full-length TRPV1-T468Tet3-Bu-GFP required incorporation of Tet3-Bu using in-gel GFP fluorescence. As shown in *Figure 3—figure supplement 1C*, detergent solubilized cell lysates from cells expressing TRPV1-T468Tet3-Bu-GFP showed a band at the expected molecular weight when the medium was supplemented with Tet3-Bu, but no GFP fluorescence at this molecular weight was observed when the medium lacked Tet3-Bu. These data indicate that TRPV1-T468Tet3-Bu-GFP incorporated Tet3-Bu without measurable incorporation of natural amino acids at the TAG codon site.

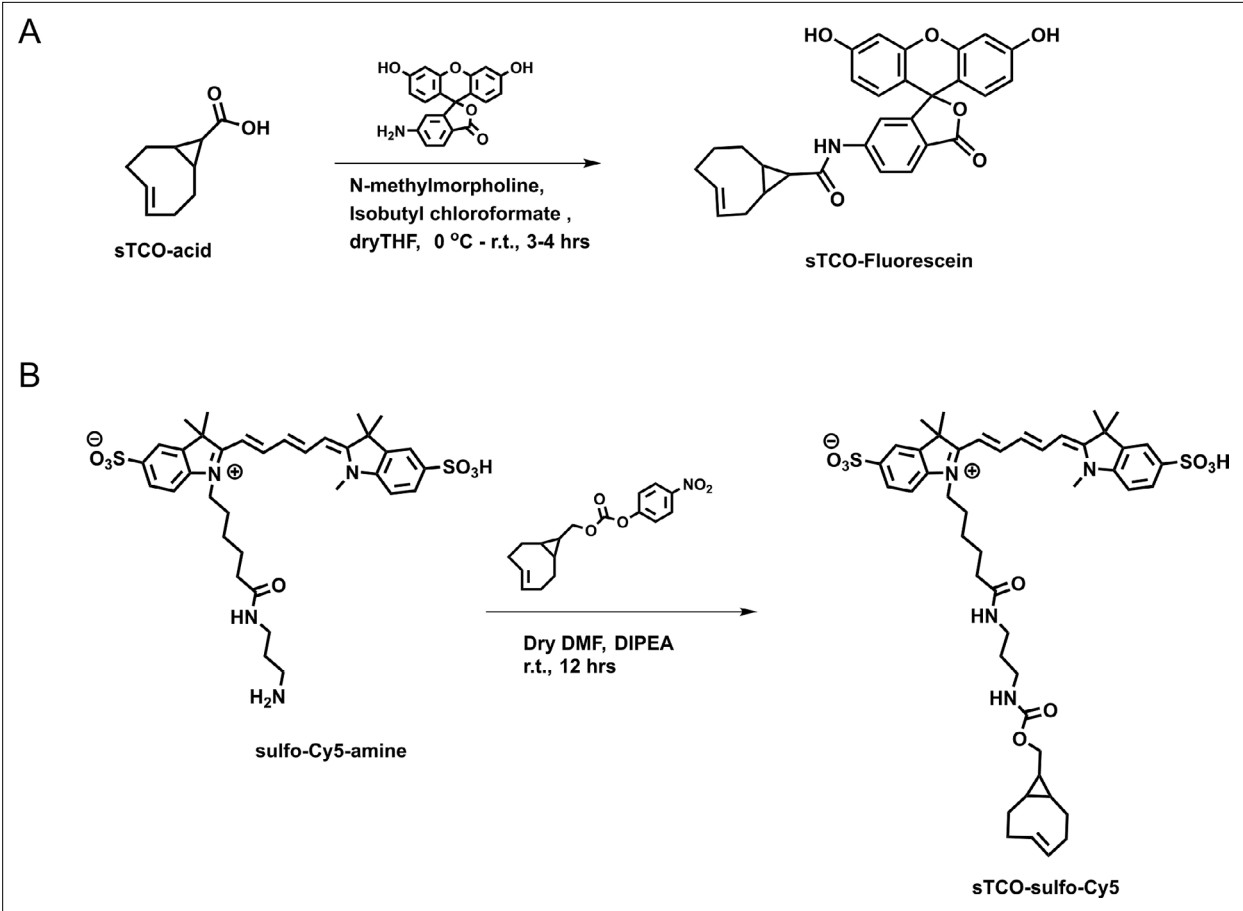

**Figure 6.** Synthesis of (**A**) sTCO-fluorescein and (**B**) sTCO-sulfo-Cy5. Schemes illustrating the synthesis pathways for the indicated dyes.

The online version of this article includes the following source data and figure supplement(s) for figure 6:

**Source data 1.** Two NMR data files processed with TopSpin 4.1.4 software.

**Figure supplement 1.** ¹H NMR spectra of synthesized compounds (**A**) sTCO-fluorescein and (**B**) sTCO-sulfo-Cy5.

## Cell culture/transfection/solutions

The experiments in *Figure 1* were performed in an NIH/3T3 cell line stably expressing PhyB-Cherry-CAAX, PIF-iSH2-YFP, and Akt-PH-CFP. These cells were a gift from Orion Weiner (UCSF, San Francisco, CA, USA). NIH-3T3 cells were cultured at 37°C, 5% $CO_2$ in Dulbecco's modified Eagle medium (Invitrogen, Grand Island, NY, USA) supplemented with 10% bovine calf serum (HyClone, Logan, UT, USA), L-glutamine (Invitrogen), and penicillin/streptomycin (Lonza, Switzerland).

F-11 cells (a gift from MC Fishman, Massachusetts General Hospital, Boston, MA, USA; *Francel et al., 1987*), a hybridoma of rat dorsal root ganglion neurons and mouse neuroblastoma cells, were used for experiments in *Figures 1 and 2*, and *Figure 3—figure supplement 2*, as they are an appropriate model for studying NGF-induced signaling in pain receptor neurons. F-11 cells were incubated in Ham's F-12 Nutrient Mixture (Gibco) supplemented with 20% fetal bovine serum, penicillin/streptomycin, and HAT supplement (100 µM sodium hypoxanthine, 400 nM aminopterin, 16 µM thymidine; Invitrogen). Click chemistry experiments in *Figure 3*; *Figure 4*; *Figure 5*; *Figure 6* were performed in HEK293T/17 cells (Catalog #CRL-11268, ATCC, Gaithersburg, MD, USA) because these cells are optimized for transfection with multiple plasmids. HEK293T/17 cells were incubated according to the manufacturer's instructions in DMEM supplemented with 10% fetal bovine serum and penicillin/streptomycin.

Cells were transfected using Lipofectamine 2000 (Invitrogen) or JetPrime (Polyplus, Illkirch, France) according to the manufacturer's instructions. Cells were passaged on to 25 mm round glass coverslips (Warner Instruments, Hamden, CT, USA) 12 hr after transfection and cultured until used for

experimentation 16–48 hr later. The coverslips were coated with poly-L-lysine to aid cell attachment (Sigma-Aldrich, St. Louis, MO, USA). For the optogenetic system, HEK293T/17 cells were transfected with PhyB-mCherry-CAAX (5 µg/well in six-well culture plate), PIF-iSH2-YFP (or PIF-YFP) 0.5 µg. The 10:1 ratio helped to express more PhyB membrane target and lesser amount of PIF cargo. When cytoplasmic PIF-iSH2-YFP was highly expressed, its translocation toward the PM upon activating 650 nm light appeared significantly reduced, likely due to the difficulty of separating signal in the cytoplasm from that at the PM. For expression of TAG-encoding cDNAs, HEK293T/17 cells in six-well plates at 30–50% confluency were transfected target gene cDNAs at 2:1 ratio (proteins with TAG codon: RS 2-84), with a total of 2 µg/well. Dominant negative elongation release factor (DN-eRF-E55D; 0.4 µg) was added to reduce truncation of the protein at the TAG codon (*Schmied et al., 2014*). For NGF stimulation, cells were transfected with 1 µg each of TrkA and p75NTR along with 1 µg TRPV1 constructs. For other constructs, 1–3 µg cDNA was used for transfection. During experiments, cells were perfused with Ringer's solution (in mM: 140 NaCl, 4 KCl, 1 $MgCl_2$, 1.8 $CaCl_2$, 10 HEPES, and 5 glucose, pH 7.3). All experiments were performed at room temperature except *Figure 3—figure supplement 2*.

## Optogenetic experiments

Cells were incubated in the dark with 10 µM PCB in the culture medium for at least 1 hr prior to experiments. To avoid photodamage of Opto3K, handling of cells during and subsequent to PCB loading was performed in the dark room using a green safety flashlight (*Levskaya et al., 2009*). Note that PhyB loaded with PCB is extremely sensitive to light, light of any wavelength between 560 and 650 nm can activate PhyB once PCB is added (*Braslavsky et al., 1997*; *Anders and Essen, 2015*). Therefore, experiments were carried out in the presence of 750 nm deactivating light and care was taken to avoid (or minimize) the exposure of cells to 560–650 nm illumination from microscopy or the external environment, including the computer monitor.

## TIRF and confocal imaging

TIRF imaging setup was as described previously (*Stratiievska et al., 2018*). We used an inverted microscope (Nikon Ti-E) equipped with a ×60 TIRF objective (NA 1.49). For activating/deactivating opto-PI3K with light, we used HQ630/20× (42490) and HQ760/40× (226723) excitation filters inserted in the overhead condenser filter slider. The whole experimental chamber was illuminated with activating or deactivating light using the maximum intensity (100 W at full spectrum) of the condenser lamp. Light intensity at the focal plane was 2.4 and 3.7 mW for 650 and 750 nm illumination, respectively.

CFP fusion proteins were imaged using excitation from a 440 nm laser and a 480/40 nm emission filter. YFP fusion proteins were monitored using the 514 nm line of an argon laser and a 525/50 nm emission filter. mCherry was excited at 561 nm and fluorescence was collected with 605/70 nm emission filter.

Time-lapse images were obtained every 10 or 20 s using either a QuantEM or an Evolve EMCCD camera (Photometrics). Movies were then processed using ImageJ software (NIH) (*Rasband, 1997*; *Schneider et al., 2012*). Regions of interest (ROI) were drawn around the footprint of individual cells and the average ROI pixel intensity was measured. Background fluorescence from cell-free areas were subtracted. Measurements were analyzed using Excel (Microsoft, Redmond, WA, USA) and Igor Pro (WaveMetrics, Portland, OR, USA). Traces were normalized by the average intensity during 1 or 3 min time period prior to activating 650 nm light.

The confocal images were obtained with a Zeiss 710 confocal microscope (Zeiss, Oberkochen, Germany). Illumination of 650–750 nm lights were controlled as described for TIRF microscopy experiments. CFP, YFP, mCherry, and JF646 were excited with 440, 514, 561, 633 nm lasers, respectively. Data were analyzed with ImageJ. The lookup table of images were changed during analysis, but the same range was applied for all images within a time series experiment.

## Electrophysiology

Currents were recorded using the standard patch clamp technique (*Hamill et al., 1981*). Borosilicate glass electrodes had a resistance of 3–6 MΩ when filled with internal solution (in mM: 140 KCl, 5 NaCl, 0.1 EGTA, and 10 HEPES, pH 7.3). Ringer's saline was used in the extracellular bath solution. Current through ion channels was measured in whole-cell configuration and the membrane potential was held at −60 mV. To activate TRPV1, the agonist capsaicin was applied for 10 s using a local perfusion system

which allowed the solution to exchange within 1 s (*Koh and Hille, 1997*). Currents were recorded with an EPC-9 amplifier (HEKA Elektronik, Lambrecht [Pfalz], Germany) and analyzed using Igor Pro software.

## Endocytosis of insulin receptor

F-11 cells expressing InsR-K676TAG-GFP were treated with 30 µM Tet3.0-Bu in the culture medium for 24 hr, transferred to glass coverslip, and further cultured in serum-deprived medium for 24 hr. The receptor targeted to the PM was labeled with membrane-impermeable sTCO-Cy3B dye (*Figure 3—figure supplement 2C*) for 4 min and free dye molecules were washed with saline solution for 2 min. Confocal images were obtained every 20 s with excitation at 561 nm and emission between 600 and 640 nm. Labeling of the receptor and insulin treatment were performed at around 35°C to promote the rate of endocytosis using a local perfusion system with heat exchanger (*Koh et al., 2011*).

## Chemical synthesis

All purchased chemicals were used without further purification. Thin-layer chromatography was performed on silica 60F-254 plates. Flash chromatographic purification was done on silica gel 60 (230–400 mesh size). $^1$H NMR spectra were recorded at Bruker 400 MHz. The chemical shifts, shown in ppm, are referenced to the residual nondeuterated solvent peak $CD_3OD$ (δ=3.31 in $^1$H NMR) as an internal standard. Splitting patterns of protons are designated as follows: s – singlet, d – doublet, t – triplet, q – quartet, quin – quintet, m – multiplet, bs – broad singlet.

sTCO-JF646: Synthesized according to the published procedure (*Jana et al., 2023*).

sTCO-fluorescein: In dry tetrahydrofuran (THF) (3 mL), sTCO-CO$_2$H (10 mg, 0.06 mmol) (*O'Brien et al., 2018*) and *N*-methylmorpholine (10 µL, 0.09 mmol) were taken under nitrogen atmosphere and stirred under ice-cold condition. Isobutyl chloroformate (10 µL, 0.07 mmol) was added dropwise to the reaction mixture and stirred for 5 min. After that, 6-aminofluorescein (24 mg, 0.07 mmol) in dry THF (1 mL) was added portion-wise and the reaction mixture was allowed to warm to room temperature and stirring was continued for another 3 hr. The solvent was evaporated, and the residue dissolved in ethyl acetate. The solution was washed with water and saturated sodium bicarbonate solution. The organic layer was dried over sodium sulfate (Na$_2$SO$_4$) and the product (16 mg, 0.032 mmol) was purified by silica gel column chromatography (10–15% methanol in dichloromethane). Yield – 53%. $^1$H NMR (400 MHz, CD$_3$OD) δ 7.91 (1H, d), 7.65 (1H, d), 7.06 (1H, d), 6.89 (1H, s), 6.84–6.82 (1H, m), 6.74–6.67 (1H, m), 6.65 (1H, s), 6.59–656 (1H, m), 6.23 (1H, d), 5.92–5.86 (1H, m), 5.27–5.20 (1H, m), 2.48 (1H, d), 2.39–2.25 (3H, m), 2.04–1.94 (2H, d), 1.37–1.32 (1H, m), 1.24 (1H, t), 1.07 (1H, t), 1.01 (1H, d), 0.84–0.75 (1H, m).

sTCO-sulfo-Cy5: In 1 mL of dry dimethylformamide (DMF), 5 mg of the sulfo-Cy5-amine (0.007 mmol) and 3.5 mg (0.01 mmol) of activated 4-nitrophenyl ester of sTCO (*Jang et al., 2020*) were added under nitrogen atmosphere. Followed by *N,N*-diisopropylethylamine (10 µL, 3 equiv.) was added to the reaction mixture and allowed to stir at room temperature for 12 hr. After that, the solvent was concentrated onto silica gel under reduced pressure and the product (4 mg, 0.004 mmol) was purified by silica gel column chromatography (20–30% methanol in dichloromethane). At the beginning of column run, added 50 mL of 50% ethyl acetate and hexane to remove the DMF solvent. Yield – 57%. $^1$H NMR (400 MHz, CD$_3$OD) δ 8.31 (2H, t), 7.97 (1H, s), 7.89 (2H, d), 7.87 (1H, dd), 7.33 (2H, d), 6.67 (1H, t), 6.32 (2H, dd), 5.87–5.79 (1H, m), 5.13–5.06 (1H, m), 4.13 (2H, t), 3.88 (2H, d), 3.72 (2H, quin.), 3.64 (3H, s), 3.21 (2H, q), 3.16 (2H, t), 3.09 (2H, t), 2.86 (4H, d), 2.29 (1H, d), 2.23–2.17 (3H, m), 1.89–1.79 (2H, m), 1.60 (2H, t), 1.38 (6H, s), 1.36 (6H, s), 0.91–0.83 (2H, m), 0.59–0.42 (2H, m), 0.45–0.37 (1H, m).

## In-gel fluorescence

HEK293T/17 cells were harvested 36–48 hr after transfection and treated with 1× SDS sample buffer (Invitrogen). Equal amounts of proteins were loaded into 3–8% Tris-acetate gradient gels and electrophoresed at 150 V for 1 hr. Fluorescence image of the gel was obtained with an Amersham ImageQuant 800 (Cytiva, Marlborough, MA, USA). GFP fluorescence was collected with excitation at 460 nm and a 525/20 nm emission filter. The gel was subsequently stained with Quick Coomassie solution and destained with distilled water for each 2 hr before image acquisition.

## Statistical analysis

Data are presented as mean ± SEM, n is the number of single cells. Student's t test was used to test the significant difference between two groups, with paired test performed as appropriate. $p < 0.05$ was regarded as significant.

## Acknowledgements

Research reported in this publication was supported in part by the National Institute of General Medical Sciences of the National Institutes of Health under award numbers R35GM145225 (to SEG) and GCE4All Biomedical Technology Development and Dissemination Center RM1GM144227 (to RAM) and the following additional awards from the National Institutes of Health: S10RR025429, P30DK017047, and P30EY001730. This was also supported in part by grants from the National Science Foundation NSF-2054824 (to RAM) and NSF-2129209 (to ENS). We thank Dr. Seung-Ryoung Jung for help with TIRF microscopy experiments.

## Additional information

### Funding

| Funder | Grant reference number | Author |
|---|---|---|
| National Institute of General Medical Sciences | R35GM145225 | Sharona E Gordon |
| National Institute of General Medical Sciences | RM1GM144227 | Ryan A Mehl |
| National Institutes of Health | S10RR025429 | Sharona E Gordon |
| National Institute of Diabetes and Digestive and Kidney Diseases | P30DK017047 | Sharona E Gordon |
| National Eye Institute | P30EY001730 | Sharona E Gordon |
| National Science Foundation | NSF-2054824 | Ryan A Mehl |
| National Science Foundation | NSF-2129209 | Eric N Senning |

The funders had no role in study design, data collection and interpretation, or the decision to submit the work for publication.

### Author contributions

Duk-Su Koh, Conceptualization, Data curation, Formal analysis, Supervision, Validation, Investigation, Visualization, Methodology, Writing – original draft, Writing – review and editing; Anastasiia Stratiievska, Conceptualization, Data curation, Formal analysis, Validation, Investigation, Visualization, Methodology, Writing – original draft, Writing – review and editing; Subhashis Jana, Resources, Formal analysis, Validation, Investigation, Visualization, Methodology, Writing – original draft, Writing – review and editing; Shauna C Otto, Validation, Visualization, Writing – review and editing; Teresa M Swanson, Validation, Investigation, Writing – review and editing; Anthony Nhim, Sara Carlson, Marium Raza, Investigation, Writing – review and editing; Ligia Araujo Naves, Conceptualization, Investigation, Methodology, Writing – review and editing; Eric N Senning, Resources, Funding acquisition, Validation, Methodology, Writing – review and editing; Ryan A Mehl, Conceptualization, Resources, Data curation, Formal analysis, Supervision, Funding acquisition, Validation, Investigation, Visualization, Methodology, Writing – original draft, Project administration, Writing – review and editing; Sharona E Gordon, Conceptualization, Data curation, Formal analysis, Supervision, Funding acquisition, Validation, Investigation, Visualization, Methodology, Writing – original draft, Project administration, Writing – review and editing

Author ORCIDs
Duk-Su Koh http://orcid.org/0000-0003-0406-2308
Ryan A Mehl https://orcid.org/0000-0003-2932-4941
Sharona E Gordon https://orcid.org/0000-0002-0914-3361

Reviewer #1 (Public Review): https://doi.org/10.7554/eLife.91012.3.sa1
Reviewer #2 (Public Review): https://doi.org/10.7554/eLife.91012.3.sa2
Reviewer #3 (Public Review): https://doi.org/10.7554/eLife.91012.3.sa3
Author response https://doi.org/10.7554/eLife.91012.3.sa4

## Additional files

### Supplementary files
• MDAR checklist

### Data availability
All source data are included in the manuscript and the supporting files.

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
