## [Editor Report · eLife assessment]

This study develops a new and **important** method for dissecting out two overlapping cell signaling pathways, phosphoinositide signaling and membrane protein trafficking. The combination of two state-of-the-art spectroscopic techniques provides **compelling** evidence for a reciprocal influence between an enzyme and a channel. The work will be of interest to the broader cell biology, biophysics and biochemistry communities.

---

## [Referee Report · Reviewer #1 (Public Review)]

Summary:

This work seeks to isolate the specific effects of phosphoinositide 3-kinase (PI3K) on the trafficking of the ion channel TRPV1, distinct from other receptor tyrosine kinase-activated effectors. It builds on earlier studies by the same group (Stein et al. 2006; Stratiievska et al. 2018), which described the regulatory relationship between PI3K, nerve growth factor (NGF), and TRPV1 trafficking. A central theme of this study is the development of methods that precisely measure the influence of PI3K on TRPV1 trafficking and vice versa. The authors employ a range of innovative methodologies to explore the dynamics between TRPV1 and PI3K trafficking.

Strengths:

A major strength of this study is the application of innovative methods to understand the interaction between PI3K and TRPV1 trafficking. The key techniques presented include:

(1) The optogenetic trafficking system based on phytochrome B, introduced in this research. Its interaction mechanism, dependent on reversible light activation, is comprehensively explained in Figures 1 and 2, with the system's efficacy demonstrated in Figure 3.

(2) An extracellular labeling method using click chemistry, which although not exclusive to this study, introduces specific reagents engineered for membrane impermeability.

The central biological insight presented here is the sufficiency of PI3K activation to guide TRPV1 trafficking to the plasma membrane. An additional notable discovery is the potential regulation of insulin receptors via this mechanism.

The paper's strengths are anchored in its innovative methodologies and the valuable collaboration between groups specializing in distinct areas of research.

Weaknesses:

The paper might benefit from a more streamlined structure and a clearer emphasis on its findings. A possible way to enhance its impact might be to focus more on its methodological aspects. The methodological facets stand out as both innovative and impactful. These experiments are well-executed and align with biological expectations. It's evident how these techniques could be tailored for many protein trafficking studies, a sentiment echoed in the manuscript (lines 287-288). When seen through a purely biological lens, some findings, like those concerning the PI3K-TRPV1 interaction, are very similar to previous work (Stratiievska et al. 2018). A biological focus demands further characterization of this interaction through mutagenesis. Also, the incorporation of insights on the insulin receptor feels somewhat tangential. A cohesive approach could be to reshape the manuscript with a primary focus on methodology, using TRPV1 and InsR as illustrative examples.

---

## [Referee Report · Reviewer #2 (Public Review)]

Summary

The authors developed new tools for isolating PI3K activity and for labeling newly made membrane proteins for monitoring membrane trafficking. They found that PI3K activity alone was able to explain the increased presence of TRPV1 on the membrane independent of other cascades induced by NGF signaling. They also showed an interesting feedback between PI3K and the insulin receptor trafficking to the membrane.

Strengths:

A major strength of the paper is the innovative combination of techniques. The first technique used the optogenetic PhyB/PIF system. They anchored PhyB to the membrane and fused PIF with the interSH2 domain from PI3K. This allowed them to use 650nm light to induce an interaction between the PhyB and PIF resulting in a recruitment of the endogenous PI3K to the membrane through the iSH2 domain without actual activation of an RTK. This allowed them to dissect out one function, just PI3K recruitment/activation from the vast number of RTK downstream cascades.

The second technique was the development of a new non-canonical amino acid that is cell-impermeant. The authors synthesized the sTSO-sulfa-Cy5 compound that will react with the Tet3 ncAA through click chemistry. They showed that the sulfa-Cy5 did not cross the membrane and would be used to track protein production over time, though the reaction rates were slow as noted by the authors. The comparison of the sulfa-Cy5 data with the standard GFP with TIRF showed a clear difference indicating the useful information that is gained with the ncAA.

Another strength comes from the discovery that an isolated PI3K is responsible for increasing TRPV1 and InR trafficking to the plasma membrane.

Weakness:

The discussion does not go into much detail regarding the importance of their discovery of TRPV1 and InR increases trafficking due to PI3K activation. It also jumps to the limitations of in vivo implementation prematurely. These weaknesses are minor however.

The authors achieved their goal of creating the tools needed to separate out one of the many RTK signals and give a strong proof of concept implementation of their tools. Their results support their conclusions and will help understand how TRPV1 is regulated by signals other than the traditional channel activators. The tools developed in the article will be of use to the broader cell biology and biophysics community, not just the channel community. The opto control of the PhyB/PIF system makes it more convenient than other systems since it does not take the typical wavelengths needed for fluorescence. The cell-impermeant ncAA will also be a great tool for those studying membrane proteins, protein trafficking and protein dynamics.

---

## [Referee Report · Reviewer #3 (Public Review)]

Summary:

In this manuscript, Koh, Stratiievska, and their colleagues investigate the mechanism by which TRPV1 channels are delivered to the plasma membrane following the activation of receptor tyrosine kinases, specifically focusing on the NGF receptor. They demonstrate that the activation of the NGF receptor's PI3K pathway alone is sufficient to increase the levels of TRPV1 at the plasma membrane.

Strengths:

The authors employ cutting-edge optogenetic, imaging, and chemical-biology techniques to achieve their research goals. They ingeniously use optogenetics to selectively activate the PI3K pathway without affecting other NGF pathways. Additionally, they develop a novel, membrane-impermeable fluorescent probe for labeling cell-surface proteins through click-chemistry.

Comment on revised version:

We commend the authors on the significant improvements made to the manuscript. They have adequately addressed our comments. Notably, the new control experiments shown in Figure 4E and Figure 5 Fig. Supp 1 convincingly demonstrate the specificity of the NGF and 650 nm light stimuli, respectively. The addition of quantitative analyses strengthens the findings significantly. Furthermore, the manuscript is now presented in a much linear manner, enhancing its clarity and impact.

---

## [Author Response]

The following is the authors’ response to the original reviews.

We thank the reviewers for their careful reading of our manuscript and their constructive comments. We have significantly improved the writing, consolidated figures, and include new experiments (see below). We now center the manuscript on the methods used and have updated the title to reflect this new emphasis. We have also added quantification with statistics, as described below. A detailed description of our improvements is provided below.

New data figures:

• Fig 3 – fig supp 2 – new experiment with insulin-triggered endocytosis of InsR

• Fig 3 – fig supp 3 – new experiments, all using the same protein construct

• Fig 3 – movie– new experiment with insulin-triggered endocytosis of InsR

• Fig 4 – added new vehicle-only negative control experiments

• Fig 5 – fig supp 1 – new negative control experiments with sequential exposures to 750 nm light

Added figure panels with quantification/statistics for: Fig. 1F; , Figure 1- figure supp 2B, Figure 2B, D, Fig. 2 – fig supp 1B, D; Fig 2 – fig supp 2B; Fig 2 – fig supp 3B;

**Reviewer #1:**
(1) The paper might benefit from a more streamlined structure and a clearer emphasis on its findings. A possible way to enhance its impact might be to focus more on its methodological aspects. The methodological facets stand out as both innovative and impactful.

We thank the reviewer for this suggestion and have rewritten the manuscript to center the methods, with our applications to TRPV1 and the InsR serving as examples.

(2) Line 243: Please provide a reference for Tet3-Bu or clarify its origin in this study. A concise description would be helpful.

The Jang et al., 2020 and Jana et al., 2023 studies are cited and give the structure of Tet3-Bu in Figure 3A.

(3) Consider merging Figures 1 and 2 for clarity.

Because the cell types and constructs expressed differ for the figures, we did not merge them. However, we moved Figure 1 to the supplement because it repeats previously published data.

(4) Lines 281 and 293 should refer to Figure 5C, not 5B.

This is now corrected.

(5) Should the paper pivot towards methodology, combining Figures 6 and 7 might be more coherent.

The experiments in Figures 6 and 7 are different, making it difficult to merge them. However, Figures 7 and 8 describe the same experimental approach applied to two different membrane proteins. To align with our new focus on the methods and deemphasis of the biological system, we have merged Figures 7 and 8.

(6) A brief discussion comparing the cell surface labeling techniques and the merits of the presented system would offer valuable context.

We agree that additional discussion here would be helpful but were also trying to satisfy Reviewer #3’s request to reduce review-like content that disrupts the flow of the primary results. We therefore did not add a discussion of cell-surface labeling techniques.

**Reviewer #2:**
(1) To monitor the phosphatidylinositol-3,4,5-trisphosphates, the pleckstrin homology (PH) domain from Akt was used. This PH domain is not specific for just PI(3,4,5)P3 as stated by the authors. The Akt PH domain also binds PI(3,4)P2. The observed PI3K localization increase will also increase PI(3,4)P2 concentrations so the observed responses may not be solely because of PI(3,4,5)P3……Repeating the PH domain experiments with a PH domain that is specific for just PI(3,4,5)P3, like GRP1 or Btk, would be useful to separate out any contributions from PI(3,4)P2.

We have repeated key experiments demonstrating optogenetic activation of PI3K with the Grp1-PH domain and included these data in Figure 1-figure supplement 2.

(2) The data in Figure 4 supplement was confusing to interpret since it is unclear whether a membrane protein with the Tet3 is being expressed at the same time as the ncAA for labeling or if the observed labeling is endogenous. If the observed labeling in Figure 4 supplement D is endogenous, then significant concerns come up regarding the background labeling of the sTCO-sulfo-Cy5 used in the rest of the experiments.

We have updated the data in this figure using the same protein (InsR-Tet3-Bu-GFP) for every sTCO-conjugated dye tested. The protein is also labelled with GFP, making it clear which cells in the field were transfected and which were not. The new panels showing the bright field images for each field further aid readers in identifying untransfected cells. We believe the new presentation addresses the reviewer’s concerns about distinguishing sTCO labeling of Tet3-Bu-incorporating protein from labeling of endogenous proteins.

(3) I recommend reorganizing the article to be more linear. For example, Figure 4 is not fully explained until after Figure 4 supplement and Figure 5. This non-linear organization required a lot of back and forth reading to fully understand the logic of the experiments as well as the conclusions.

We have improved the presentation along the lines suggested by the reviewer.

(4) The InsR data is interesting as a proof of concept however the writing around the InsR looks like an afterthought. The explanation for why InsR is chosen, what is known and unknown about its trafficking is given secondary importance in the writing but not in the figures. This difference weakens the article.

We have improved the presentation along the lines suggested by the reviewer.

(5) Line 244 should read Figure 4A.

This is now corrected.

(6) Line 281 should read Figure 5C.

This is now corrected.

(7) Line 645. Fig 4, says C and E were shown as inverted b&w images when they aren't.

This is now corrected.

(8) Fig 8. Line 702. States that these are TRPV1 positive cells but the figure is about InsR.

This is now corrected.

**Reviewer #3:**
(1) The Results section is lengthy and disorganized. Consider revising it for better clarity and conciseness. For instance, moving lines 157 and 166-170 to the Discussion or Methods section can streamline the Results section.

We have improved the presentation along the lines suggested by the reviewer.

(2) Provide more specificity in reporting: In lines 139-170, clarify why you chose to use PhyB and this particular technique. Eliminate extraneous details and maintain a more concise narrative.

We have improved the presentation along the lines suggested by the reviewer.

(3) Avoid excessive review-like content, and keep the Results section focused on presenting novel findings. Simplify lines 4 173-185 to provide a straightforward presentation of results rather than extensive references to previous work.

We have improved the presentation along the lines suggested by the reviewer.

(4) Reevaluate lines 196-204 to determine if they are best suited for the Results section or if they could be moved to the Discussion or Methods for improved focus.

We have improved the presentation along the lines suggested by the reviewer.

(5) 231-238, revise the content to be more concise and directly to the point.

We have improved the presentation along the lines suggested by the reviewer.

(6) Limit the number of figures to a maximum of five and restructure them to enhance readability. Consider consolidating panels from Figures 1 (which replicates previouslypublished work), 2, and 3 into a single figure to improve organization and information flow.

See response to Reviewer #1, Comment #3. Although we did not merge Figures 2 and 3, we have consolidated the writing to improve the flow of the writing.

(7) Move Fig 5, which depicts control experiments, to supplementary information to improve the overall flow of the paper. Also, Figure 5 comes in the text before Figure 4 C-F and before Figure 4- supp1, so placing it in supplementary information would fix this issue.

We have moved this figure to the supplement as Figure 3 – figure supplement 1.

(8) Merge Figures 6, 7, and 8 (or at least 7 and 8) to facilitate the comparison of data obtained with different proteins or conditions.

We have merged Figures 7 and 8.

(9) Line 303: when referring to the chemical structure of sTCO-sulfo-Cy5, refer to Figure 4 Supp 1 and not Figure 9. Alternatively, consider moving Fig 9 to supplementary information or placing it earlier in the figure list.

We now refer to the earlier supplemental figure when describing the structure of sTCO-sulfo-Cy5.

(10) Ensure proper referencing of Figure 4E in the text, particularly since it's vital to understanding the selection of mutation sites for the Insulin receptor, as discussed in lines 392-400.

We have made this correction.

(11) Maintain citation consistency by verifying that all references cited in the text, including those in the Introduction, Results, and Discussion sections, are included in the References list at the end of the paper.

We have reviewed all our citations for consistency.

The reviewer is also concerned by the lack of any statistical analyses, and of appropriate control experiments:(1) The trapping of PI3K at the plasma membrane, shown in Figure 3 supplementary 1, is not very convincing. It is unclear whether PI3K is trapped at the membrane, as claimed by the authors, or whether PI3K slowly accumulates at the membrane independently of the light stimulation. Indeed, the baseline fluorescence isn't flat to start with (especially in F-11 cells), and the change in fluorescence under 650 nm light is very modest, much weaker, in fact, than in control experiments without TRPV1 (Figure 2C). Do the authors observe a similar drift in fluorescence in absence of photostimulation at 650 nm? Such control experiment needs to be performed and discussed. More importantly, authors need to provide quantitative (and not just qualitative) measures of the changes in fluorescence observed in the different conditions, and run adequate statistical analyses to compare the different conditions (for all the figures of the manuscript where this applies).

We can see that the language of “trapped at the membrane” is more of an interpretation than a description. We now describe this result as a lack of dissociation of PIF-iSH2 from the membrane in response to 750 nm light. We more clearly explain our interpretation and label it as speculative.

(2) Consider moving Figure 3 Supplementary 1 from supplementary information to the main document due to its importance. It seems like an important finding to me, and I believe also to the authors, who wrote a whole paragraph on PI3K trapping in the discussion section (lines 361-380).

We agree that the results from this figure are important. To better align with the request of all reviewers to shorten the manuscript and reduce the number of figures in the main text, however, we have left the figure in the supplement.

(3) Figure 3: why is the increase in IP3 levels not reversible as in Figure 2? Is this because IP3 is detected only at the membrane level (TIRF experiment) and not the entire cell? Authors should comment on this aspect.

As described in response to Comment#2, we now better explain our interpretation. Briefly, we speculate that the PIF-iSH2 that encounters TRPV1 in the plasma membrane binds to the ankyrin repeat domain of TRPV1 and, therefore, does not readily dissociate from membrane in response to 750 nm light.

(4) Figure 4E: Verify the functionality of the Insulin receptor mutants, as was done for TRPV1.

We have added new experiments to demonstrate that the insulin receptor incorporating Tet3-Bu is functional. Because the insulin receptor is not electrogenic, we could not use electrophysiology to validate its function. Instead, we measured the insulin-dependent endocytosis of the receptor. These data are now presented in Figure 3 – figure supplement 2 and Figure 3 – supplemental movie.

(5) Figures 6 to 8: The authors quantify the change in plasma membrane expression of TRPV1 and insulin receptors after NGF treatment (or photoactivation), but an important control experiment is missing. They first label cells with sulfo-Cy5, then treat them with NGF (or photoactivate them with 650 nm light), and then label them again with sulfo-Cy5, supposedly to label only the TRPV1 receptors that newly arrived at the membrane. However, we have no evidence that the first sulfo-Cy5 labeling (1 uM, 5 min) was complete. In fact, labeling with sulfo-Cy5 (200 nM) in Figure 4 never reaches saturation, not even after 20 min. The authors need to control for this, by comparing the change in fluorescence with and without NGF treatment. The GFP control is simply not sufficient. Also, include Figure 8 in the text, as it is missing from the results section, and discuss the results in more detail. Indeed, the current data is appealing as it suggests that what was observed with TRPV1 is also true for the Insulin receptor, but without a proper control this could just be an artefact.

We have performed several new control experiments to address the reviewer’s concerns. (1) For NGF-induced increase in TRPV1 at the plasma membrane, we repeated the experiment using a vehicle instead of NGF. These data, added to Figure 4E, demonstrate that the increase in plasma membrane TRPV1 depends on NGF. (2) For the light-activated increase in plasma membrane TRPV1, we repeated the experiment using a second exposure to the deactivating 750 nm light instead of the activating 650 nm light and added the data as Figure 5, figure supplement 1A-E. These new data demonstrate that the increase in plasma membrane TRPV1 occurred only in response to the activating wavelength of light. (3) To address the same as the previous comment, but for the insulin receptor, we repeated the insulin receptor experiments also using a second exposure to the deactivating wavelength of light. These data are now shown in Figure 5, figure supplement 1F-I and demonstrate that the increase in the insulin receptor levels in the plasma membrane required the activating wavelength of light.

(6) Line 313: "Importantly, sTCO-sulfo-Cy5 did not appear to equilibrate across the cell membrane and did not label untransfected cells (i.e., those without GFP; Figure 4 - figure supplement 1)". I don't see where the absence of labeling of untransfected cells is shown. The authors should show fluorescence changes on the surface of both transfected and untransfected cells and, as discussed above, quantify the data and provide statistical analyses.

See response to Reviewer #2, Comment #2.

Minor Comments:(1) Define « PM » and « RTK » in abstract We have made the requested changes.(2) Consider presenting the signaling pathways defined in the introduction in a scheme to improve readability.

We have added the signaling pathways defined in the introduction to Figure 1A.

(3) In Figure 1A, include the CAAX lipidation signal in the schematic representation.

We had already shown the lipidation itself, but we have added the lipidation signal as a magenta star, with its meaning explained in the figure legend. We hope the reviewer finds this useful.

(4) Terminology clarification: Given the broad readership of Elife, provide clearer explanations for terms and techniques used, such as the function of PIF (line 144).

We define the acronym PIF in the text, but do not further elaborate on the biological function of PIF to align with other reviewers’ requests that we reduce the review-type material in the manuscript.

(5) Correct "m-1s-1" to "M-1s-1" in line 119.

This is now corrected.

(6) Replace "activate" with "activation" in line 122.

This is now corrected.

(7) Indicate 650 nm and 750 nm next to the arrows in Figure 2B for reader clarity.

We have added the requested arrow labels.

(8) Correct Figure 5A to Figure 4A in line 244.

This is now corrected.

(9) Correct Figure 5B to Figure 5C in line 293.

This is now corrected.

(10) In lines 274, 293, 312 and 329, clearly specify which panels of the referenced figures are being discussed to avoid confusion.

We have now clearly specified which panels are being referenced.

(11) Figure 1B: it is unclear how long after 650 nm light switching the image is taken. The red bar indicating 650 nm light makes it look like the image is taken right after light switching, which would suggest that PIF-YFP trafficking to the membrane takes milliseconds in response to 650 nm light. However, the legend says that photoactivation kinetics are in the range of 10 seconds. Please accurately position the red bar in Figure 1B to reflect the time between light switching and imaging, and specify the time between light switching and imaging in the figure legend.

We have more accurately shown the timing of image acquisition in what is now Figure 1, figure supplement 1.

(12) Please add a merged image for all the immune data figure.

We are uncertain about which figures the reviewer is referring to. We do not have any immunohistochemistry in the manuscript.

(13) Line 205: "we found that expression of TRPV1 trapped PIF-iSH2 at the PM upon stimulation with 650 nm light, so that it no longer translocated to the cytoplasm in response to 750 nm light (Figure 3B and Figure 3 - figure supplement 1A)." This is shown in the supplementary figure but not in Figure 3B. Same issue with the following sentence.

We have corrected the figure references in the text.

(14) For Figures 7 and 8, the authors state ""We next asked whether click chemistry labeling could be executed in cells in which we also used the PhyB/PIF machinery for activating PI3K." Is this really the main motivation for conducting these experiments?

Good point. We have improved the writing around this issue.